# Towards Precise Prediction Uncertainty in GNNs: Refining GNNs with Topology-grouping Strategy

## Abstract

The calibration of model predictions has recently gained increasing attention in the domain of graph neural networks (GNNs), with a particular emphasis on the underconfidence exhibited by these networks. Among the critical factors identified to be associated with GNN calibration, the concept of neighborhood prediction similarity has been recognized as a pivotal component. Building upon this insight, modern GNN calibration techniques adapt node-wise temperature scaling by smoothing the confidence of individual nodes with those of adjacent nodes. However, these approaches often engage in superficial learning across varying affinity levels, thereby failing to effectively accommodate diverse local topologies. Through an in-depth analysis, we unveil that calibrated logits from preceding research significantly contradict their foundational assumption of nearby affinity, necessitating a re-evaluation of the existing GNN-founded calibration strategies. To address this, we introduce SIMI-MAILBOX, which categorizes nodes based on both neighborhood representational similarity and their own confidence, irrespective of proximity or connectivity. Our method effectively mitigates miscalibration for nodes exhibiting analogous similarity levels by adjusting their predictions with *group-specific* temperatures. This encourages a more sophisticated calibration, where each group-wise temperature is tailored to address affiliated nodes with similar topology. Extensive experiments demonstrate the effectiveness of SIMI-MAILBOX across diverse datasets on different GNN architectures.

## 1 Introduction

Graph-structured data have been extensively employed to represent various types of data, including molecular arrangements (Gilmer et al., 2017), social networks (Hamilton et al., 2017), and e-commerce transaction histories (Wu et al., 2019). Emerging as a powerful tool in this arena, Graph Neural Networks (GNNs) have achieved impressive performance on modeling graph data and addressing diverse graph-based tasks, such as node classification (Kipf & Welling, 2016; Hamilton et al., 2017; Xu et al., 2018; Wang et al., 2020; Park et al., 2021), link prediction (Zhang & Chen, 2018; Li et al., 2018; Yun et al., 2021; Ahn & Kim, 2021; Zhu et al., 2021), and graph classification (Hamilton et al., 2017; Xu et al., 2018; Lee et al., 2018; Sui et al., 2022; Hou et al., 2022).

Beyond achieving correct prediction, the precise quantification of prediction uncertainty is nontrivial for the reliable utilization of neural networks in downstream decision-making process. Acknowledging the importance of reliable prediction, numerous calibration studies have been actively proposed on image and language domains (Guo et al., 2017; Mukhoti et al., 2020; Zhang et al., 2020; Xing et al., 2019; Jiang et al., 2021; Minderer et al., 2021; Gruber & Buettner, 2022). Recently, network calibration has also drawn attention in the field of GNNs (Wang et al., 2021b; Hsu et al., 2022b;a; Shi et al., 2022; Wang et al., 2022; Liu et al., 2022), revealing a tendency of GNNs to exhibit underconfident predictions. Building upon this insight, contemporary studies in GNN calibration, CaGCN (Wang et al., 2021b) and GATS (Hsu et al., 2022b), have devised node-wise temperature scaling techniques to encourage confidence adjustment between adjacent nodes. Among the crucial factors identified to be associated with miscalibration in GNNs, the concept of neighborhood prediction similarity has emerged as a fundamental component in these studies. Specifically, CaGCN asserts that nodes with

disparate neighbors should ideally possess lower confidence levels compared to the opposite case, owing to the difficulties inherent in accurately classifying such instances within the local message propagation in GNNs. Moreover, GATS elucidates the correlation between neighborhood prediction similarity and calibration errors, indicating the highest errors for nodes with disagreeing neighbors.

However, in our comprehensive analysis, we identify a notable inconsistency between their foundational assumptions of neighborhood affinity and empirical calibration results across diverse benchmark datasets. Our investigation reveals that both methods often fail to comprehensively refine predictions across varying levels of neighborhood similarity, inducing a superficial learning. For instance, CaGCN fails to refine the confidence of nodes with low similarity, from which they should elevate to reach the desired accuracy, breaking down their ground principles. Furthermore, GATS demonstrates sub-optimal calibration in regions comprising nodes with low similarity, where they have revealed that large miscalibration occurs. This invokes a re-evaluation of the adequacy of existing GNN-based calibration strategies in reflecting local similarity characteristics.

To address these inconsistencies, we introduce a novel calibration method specifically devised to rectify the identified limitations in a post-hoc fashion. Named as SIMI-MAILBOX, our method categorizes nodes based on neighborhood representational similarity and confidence, irrespective of proximity or connectivity. This process is rooted in our empirical validation that nodes displaying analogous neighborhood affinity and confidence levels exhibit a similar degree of miscalibration. Following this, our SIMI-MAILBOX allocates *group-specific* temperatures to adjust predictions of analogous nodes within each group. Through this confidence refinement with per-group specialized temperatures, our method effectively mitigates miscalibration across varying local topologies.

Our contributions are summarized in three-fold:

- We elucidate the limitations inherent in current calibration methods, especially concerning neighborhood prediction affinity - a recognized key component for GNN calibration.
- Given the limitations observed in preceding research, we introduce SIMI-MAILBOX, which classifies nodes according to similar local topology and confidence, rectifying the miscalibration via per-group specific temperatures.
- We validate the proposed method through comprehensive experiments, incorporating both quantitative and qualitative evaluations of calibration performance.

Overall, our contributions provide valuable insights into the calibration of GNNs and propose an effective calibration method, capable of enhancing the reliability of GNN predictions.

## 2 RELATED WORKS

**Post-hoc Calibration Methods.** Network calibration has been a popular research topic in image and language domains, with various methods aiming at enhancing the calibration of neural networks (Guo et al., 2017; Kull et al., 2017; 2019; Wang et al., 2021a; Mukhoti et al., 2020; Rahimi et al., 2020; Han et al., 2021; Kumar et al., 2018; Gupta et al., 2020; Xing et al., 2019; Jiang et al., 2021; Yu et al., 2022; Widmann et al., 2019; Wald et al., 2021). Among these diverse approaches, post-hoc calibration methods have found widespread adoption, owing to their straightforward application and strong performance. Considerable techniques have focused on capturing data-specific characteristics through adjustable parameters, such as Platt scaling (Platt et al., 1999), Temperature scaling (TS) (Guo et al., 2017), and Ensemble temperature scaling (ETS) (Zhang et al., 2020). Above all, TS has emerged as a prevalent choice due to its simplicity, by offering an extension of Platt scaling to facilitate multi-class calibration. This method aligns model predictions with accuracy by leveraging a single temperature parameter that modulates the uncalibrated logits. Apart from standard Expected Calibration Error (ECE) metric (Naeini et al., 2015), some commonly used metrics for evaluating the calibration performance of neural networks are class-wise ECE (Kull et al., 2019; Nixon et al., 2019) Brier score (Brier et al., 1950), negative log-likelihood, and Kernel Density Estimation-based ECE (KDE-ECE) (Zhang et al., 2020).

**Grouping-based Calibration Works.** Addressing miscalibrations in a group-wise manner has been studied in (Hébert-Johnson et al., 2018; Yang et al., 2023; Perez-Lebel et al., 2022). Hébert-Johnson et al. (2018) introduced multicalibration strategy, aiming to achieve calibration within diverse,

overlapping subgroups to enhance both fairness and accuracy in machine learning models. Yang et al. (2023) proposed a new semantic partitioning approach for neural network calibration and utilized learnable grouping function to refine calibration beyond traditional methods. Meanwhile, Perez-Lebel et al. (2022) presented the concept of 'grouping loss' as a novel metric to assess the variance in true probabilities sharing the same confidence score, challenging traditional calibration approaches. Nevertheless, our approach exhibits a clear fundamental difference from these works, which will be discussed in Appendix B.

**Calibration methods for Graph Neural Networks.** Recent literature has highlighted an increased focus on the calibration of Graph Neural Networks (GNNs) with post-processing calibration strategies (Wang et al., 2021b; Hsu et al., 2022b;a; Wang et al., 2022; Shi et al., 2022; Liu et al., 2022). A pioneering study, Wang et al. (2021b) discovered that GNNs exhibit an underconfident behavior, which contradicts the prevailing notion that Deep Neural Networks (DNNs) possess overconfidence. They proposed CaGCN that leverages GCN as a calibration function to produce unique temperature values for each node based on the predictions of its adjacent nodes. (Hsu et al., 2022b) extended the knowledge of GNN calibration dynamics by identifying several factors that foster GNN calibration errors. Based on these factors, they designed a Graph Attention Network (GAT)-based calibration function, GATS, that generates distinct temperature values for individual nodes considering these factors. In (Hsu et al., 2022a), they also proposed an edge-wise expected calibration error metric that accounts for the non-iid nature of graph structures, which is overlooked in traditional node-wise ECE. Furthermore, they subdivided the metric into homophilous and heterophilous cases, respectively referred to as agree-ECE and disagree-ECE. Diverging from these approaches, GCL (Wang et al., 2022) addressed the underconfidence observed in GNNs through the incorporation of a minimal-entropy regularization component into the cross-entropy loss function, promoting an up-weighting of the loss attributed to nodes exhibiting high confidence levels.

## 3 PRELIMINARIES

**Problem Setup.** We focus on calibrating GNNs for semi-supervised node classification in a post-hoc context. Let an undirected graph be denoted as $\mathcal{G}(\mathcal{V}, \mathcal{E})$, where $\mathcal{V}$ and $\mathcal{E}$ indicate the sets of vertices and edges respectively. The vertex set $\mathcal{V}$ is represented by a feature matrix $\boldsymbol{X} = [\mathbf{x}_1^\mathsf{T}, ..., \mathbf{x}_{|\mathcal{V}|}^\mathsf{T}] \in \mathbb{R}^{|\mathcal{V}| \times D}$ and the edge set $\mathcal{E}$ is denoted by an adjacency matrix $\boldsymbol{A} \in \mathbb{R}^{|\mathcal{V}| \times |\mathcal{V}|}$. Given the node-wise predictions $\hat{y} = [\hat{y}_1, ..., \hat{y}_{|\mathcal{V}|}]^\mathsf{T}$ and output confidence $\hat{p} = [\hat{p}_1, ..., \hat{p}_{|\mathcal{V}|}]^\mathsf{T} \in \mathbb{R}^{|\mathcal{V}|}$ from a trained GNN, the GNN $f_\theta$ is *well-calibrated* if $\hat{p}_i$ for each node $i$ accurately serves the ground-truth probability $p_{\text{true}}$, formulated as below:

$$\mathbb{P}(\hat{y}_i = y_i | \hat{p}_i = p_{\text{true}}) = p_{\text{true}}, \quad \forall p_{\text{true}} \in [0, 1]. \tag{1}$$

The expected calibration error (ECE) (Naeini et al., 2015) has been recognized as the de facto metric to evaluate the calibration quality of network predictions. ECE groups nodes according to their confidences into $M$ equally partitioned confidence intervals $\{B_1, ..., B_M\}$ and assesses the expected discrepancy between accuracy and average confidence within individual bins:

$$\text{ECE} = \sum_{m=1}^{M} \frac{|B_m|}{|\mathcal{V}|} \Big| \text{acc}(B_m) - \text{conf}(B_m) \Big|, \tag{2}$$

where $|B_m|$ refers to the number of nodes within the $m$-th interval. Here, per-bin accuracy and average confidence for the $m$-th bin are defined as $\text{acc}(B_m) = \frac{1}{|B_m|} \sum_{i \in B_m} \mathbf{1}[y_i = \hat{y}_i]$ and $\text{conf}(B_m) = \frac{1}{|B_m|} \sum_{i \in B_m} \hat{p}_i$, respectively.

**Exploration of Neighborhood Similarity in Prior Studies.** The concept of neighborhood similarity has been recognized as a primary element in the field of GNN calibration (Wang et al., 2021b; Hsu et al., 2022b;a; Liu et al., 2022). Among them, CaGCN (Wang et al., 2021b) advocates that given the challenges GNNs encounter in accurately classifying nodes with conflicting neighbors, the confidence levels in such cases should ideally remain still or decrease. Conversely, confidence for nodes linked to agreeing nodes should elevate, addressing the prevalent underconfidence in GNNs. Stemmed from this insight, they employ GCN (Kipf & Welling, 2016) as a calibration function to propagate the predictions to neighboring counterparts, assuming that confidence of adjacent nodes given by well-calibrated models should be analogous to each other.

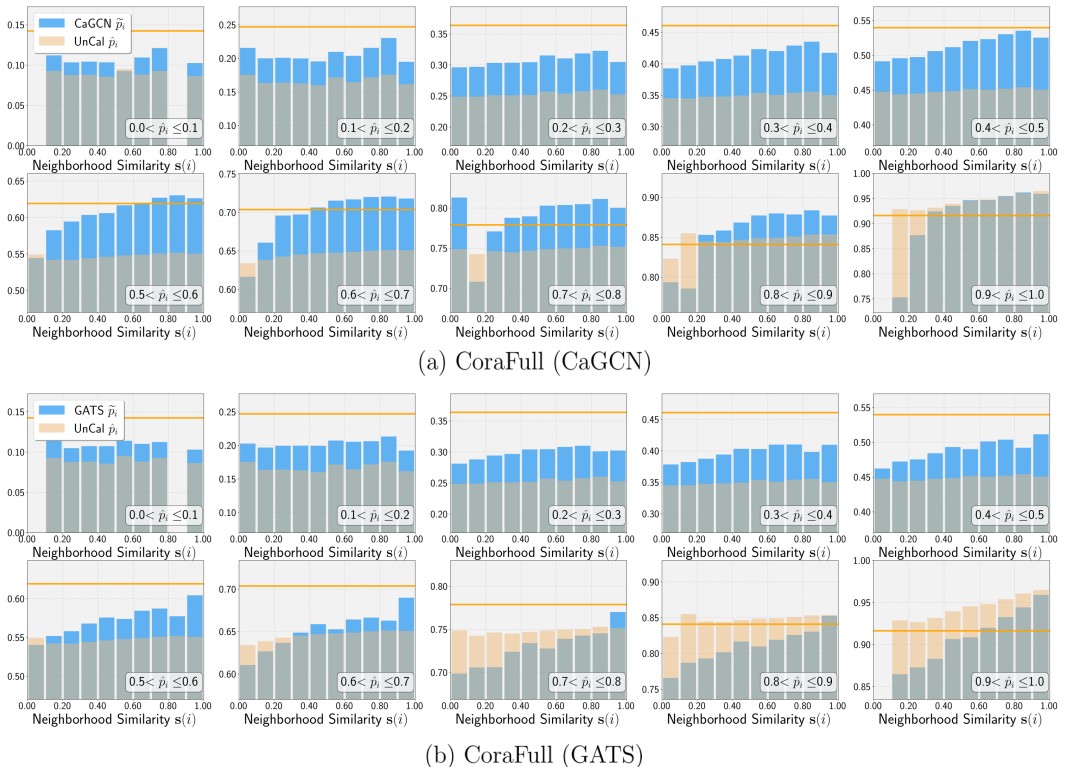

(a) CoraFull (CaGCN)

(b) CoraFull (GATS)

Figure 1: Investigation results of calibrated logits via CaGCN and GATS. **Orange** line represents *per-confidence* bin accuracy, while **yellow** and **blue** bars denote the average confidence of uncalibrated and calibrated logits in each *neighborhood similarity* sub-interval, respectively. Note that the **gray** area indicates the intersection between two bars. As illustrated, the confidence of sub-intervals with low similarity where CaGCN have advocated to remain still or decrease, should increase to meet the desired accuracy. On the other hand, GATS exhibits sub-optimal calibration on these sub-intervals, from which they discovered to reveal the highest calibration error.

In parallel, GATS (Hsu et al., 2022b) underscores the correlation between neighborhood prediction similarity and calibration error, demonstrating an increment in error with a decrement in similarity. This intimate relationship is incorporated into the normalized attention coefficients within their GAT (Veličković et al., 2017)-founded temperature function.

Denoting the calibrated confidence from the original logit $z_i$ and softmax operation as $\widetilde{p}_i$ and $\sigma_{\mathrm{sm}}$, their node-wise temperature scaling procedure is formulated as follows:

$$\widetilde{p}_i = \max_k \sigma_{\mathrm{sm}} \left( \frac{z_i}{\boldsymbol{T}_i} \right)_k .\tag{3}$$

## 4  IN-DEPTH ANALYSIS: SURFACE LEARNING ON DISTINCT SIMILARITY LEVELS

In this section, we provide a thorough investigation of calibrated predictions from earlier research across various neighborhood similarity levels. Leveraging GCN as a backbone architecture, we initially conduct equal-width binning on the confidence of all uncalibrated predictions for CoraFull dataset. Within each confidence interval, nodes are categorized based on the neighborhood prediction similarity, $\boldsymbol{s}(i)$, which is defined as the portion of neighbors assigned to the same predicted labels:

$$\boldsymbol{s}(i) = \frac{\sum_{j \in \mathcal{N}_i} \mathbb{1}[\hat{y}_i = \hat{y}_j]}{|\mathcal{N}_i|},\tag{4}$$

where $\mathcal{N}_i$ serves as a set of neighbors associated with node $i$. Subsequently, we draw the actual accuracy for each confidence bin (represented as **orange** horizontal lines) and the average of both the

Table 1: Variance of calibration errors ($\times \mathbf{100}$) involving neighborhood similarity sub-intervals (Neig. Sim.), confidence intervals (Conf), and total nodes (Node-wise).

| GNNs | | Cora | Citeseer | Pubmed | Computers | Photo | CS | Physics | CoraFull |
|---|---|---|---|---|---|---|---|---|---|
| GCN | Node-wise | 6.139 | 1.957 | 1.370 | 40.370 | 7.200 | 31.470 | 3.312 | 34.560 |
| | Conf. | 0.065 | 0.060 | 0.068 | **0.060** | 0.052 | 0.058 | 0.041 | 0.064 |
| | Neig. Sim. | **0.057** | **0.046** | **0.061** | 0.062 | **0.047** | **0.052** | **0.040** | **0.057** |
| GAT | Node-wise | 8.656 | 2.570 | 1.614 | 44.980 | 14.550 | 42.660 | 3.346 | 58.67 |
| | Conf. | 0.068 | 0.062 | 0.068 | 0.048 | 0.053 | 0.050 | **0.035** | 0.062 |
| | Neig. Sim. | **0.056** | **0.044** | **0.055** | **0.045** | **0.047** | **0.044** | **0.035** | **0.055** |

nodes' original confidence (shown as **yellow** bars) and their calibrated confidence (shown as **blue** bars) within each neighborhood affinity sub-interval, as depicted in Figure 1. The **gray** region represents the intersection between the average confidence of calibrated and uncalibrated bars. Ideally, the confidence for each neighborhood affinity subgroup should perfectly align with the actual accuracy, touching the orange horizontal line.

Notably, our findings illustrate that both CaGCN and GATS exhibit significantly different behaviors from their motivations. To be specific, CaGCN fails to adjust the confidence of sub-intervals with low similarity, where their confidence in fact should elevate to achieve the desired accuracy. This is especially observed in the $\hat{p}_i \in (0.9, 1.0]$ confidence interval, where the maximum discrepancy between accuracy and calibrated confidence in $\boldsymbol{s}(i) \in (0.1, 0.2]$ reveals approximately $16.34\%$. This highlights a counter-intuitive divergence from their foundational assumption that confidence surrounded by disagreeing counterparts should either maintain or decrease. Moreover, GATS demonstrates sub-optimal calibration in areas of low prediction similarity, from which they have contended the emergence of high calibration errors. Such a pattern is prevalent across all confidence intervals, particularly pronounced in the $\hat{p}_i \in (0.2, 0.4]$ and $\hat{p}_i \in (0.6, 0.8]$ ranges, where the average discrepancies are $7.45\%$ and $7.17\%$ in $\boldsymbol{s}(i) \in (0, 0.4]$, respectively. Hence, our observations raise a concern that existing calibration strategies are potentially inadequate in reflecting neighborhood topological similarity, a central element for calibrating GNNs. Similar investigation results on other benchmark datasets with GCN and GAT are provided in Appendix C.

## 5 PROPOSED METHOD

Given the limitation of earlier studies in Section 4, we introduce SIMI-MAILBOX, a novel post-hoc calibration technique designed to refine miscalibration in GNNs across varying levels of neighborhood similarity. Leveraging our novel observation on the importance of analogous topology categorization, SIMI-MAILBOX classifies nodes based on neighborhood representational similarity and their own confidence. Following this, our method scales the predictions of nodes within the same group via group-specific temperatures, encouraging a more sophisticated rectification.

### 5.1 INTUITION: TOPOLOGY GROUPING MATTERS

We present a novel observation accentuating that nodes with analogous neighborhood affinity and confidence exhibit similar calibration errors. For a thorough comparison, we evaluate the variance of calibration errors under three distinct circumstances: (1) Node-wise variance involving all nodes (specified as **Node-wise**), (2) variance within each confidence interval (specified as **Conf.**), and (3) variance within each neighborhood similarity sub-bin located within each confidence interval (specified as **Neig. Sim.**).

To examine the third scenario, we assess the variability in calibration error across neighborhood similarity sub-intervals , $B^{\text{sim}}$, within each distinct confidence interval, $B^{\text{conf}}$. Initially, the variance of calibration error in each $B^{\text{sim}}$, denoted as $V(B^{\text{sim}})$, is calculated as follows:

$$V(B^{\text{sim}}) = \frac{1}{|B^{\text{sim}}| - 1} \sum_{i \in B^{\text{sim}}} (D_i - \bar{D}^{\text{sim}})^2,$$

$$D_i = |\text{Acc}(B^{\text{conf}}) - \hat{p}_i|, \quad i \in B^{\text{conf}}.$$

(5)

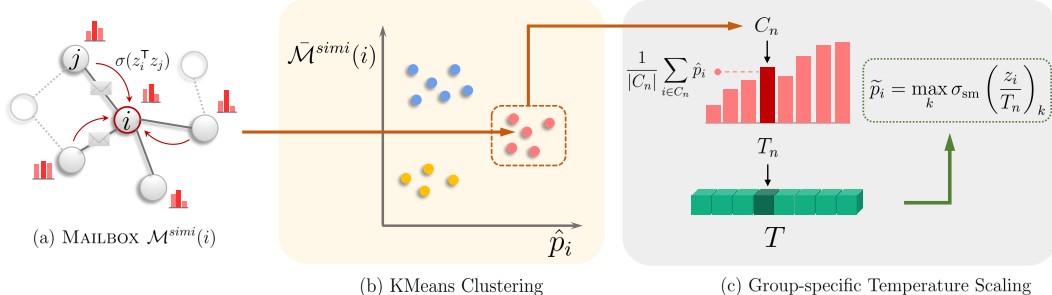

(a) Mailbox $\mathcal{M}^{simi}(i)$

(b) KMeans Clustering

(c) Group-specific Temperature Scaling

Figure 2: Overall framework of our SIMI-MAILBOX. Motivated by our novel discovery discussed in Section 5.1, we classify nodes based on analogous neighborhood prediction similarity and their own confidence, which is achieved via KMeans clustering. Subsequently, analogous nodes within each cluster is rectified via group-specific temperatures, leading to sophisticated refinement.

Here, $D_i$ is the absolute discrepancy between the accuracy associated with $B^{\mathrm{conf}}$ and confidence of individual nodes $i$ in $B^{\mathrm{conf}}$. Meanwhile, $\bar{D}^{\mathrm{sim}}$ indicates the mean discrepancy over nodes in $B^{\mathrm{sim}}$. Following this, we average the whole calculated variances over the collection $S$, which incorporates all $B^{\mathrm{sim}}$ spanning the entire $B^{\mathrm{conf}}$:

$$\bar{V}^{\mathrm{sim}} = \frac{1}{|S|} \sum_{B^{\mathrm{sim}} \in S} V(B^{\mathrm{sim}}). \tag{6}$$

Correspondingly, we compute the variability in calibration error related to the confidence interval (the second scenario) as the average variance across all $B^{\mathrm{conf}}$, considering individual variance $V(B^{\mathrm{conf}})$:

$$V(B^{\mathrm{conf}}) = \frac{1}{|B^{\mathrm{conf}}| - 1} \sum_{i \in B^{\mathrm{conf}}} (D_i - \bar{D}^{\mathrm{conf}})^2, \tag{7}$$

where $\bar{D}^{\mathrm{conf}}$ refers to the average discrepance over nodes in $B^{\mathrm{conf}}$. Following the approach of GATS, node-wise calibration error is conceptualized as the calibration error of the confidence interval that each node is associated with. Consequently, the variance in calibration error related to individual nodes (the first scenario) is defined as the variance of all node-wise calibration errors.

As outlined in Table 1, the variance of nodes within neighborhood similarity sub-intervals (**Neig. Sim.**) reveals the lowest variance, particularly when compared against the variance involving all nodes (**Node-wise**). This substantiates the notion that nodes exhibiting similar neighborhood topology and confidence levels incur analogous miscalibrations.

## 5.2 SIMI-MAILBOX: A TOPOLOGY-GROUPING STRATEGY FOR REFINING GNNS

Building on the observation discussed in Section 5.1, our SIMI-MAILBOX categorizes nodes by considering both neighborhood representational similarity and confidence levels. Our method begins by computing the neighborhood affinity, termed as MAILBOX $\mathcal{M}^{simi}(i)$, for each node $i$ through averaging representational similarity with its neighboring counterparts, defined as:

$$\mathcal{M}^{simi}(i) = \frac{1}{|\mathcal{N}_i|} \sum_{j \in \mathcal{N}_i} \sigma(z_i^{\mathsf{T}} z_j), \tag{8}$$

where $z_i$ corresponds to the output logits for node $i$ from the trained GNN, and $\sigma$ indicates a sigmoid function. Subsequently, nodes exhibiting analogous MAILBOX values and confidence levels are classified into $N$ distinct groups. More precisely, SIMI-MAILBOX constructs a feature vector $F_i^{simi} = [\hat{p}_i, \bar{\mathcal{M}}^{simi}(i)]^{\mathsf{T}}$ for each node $i$, with the first dimension representing confidence $\hat{p}_i$ and the second dimension representing a normalized MAILBOX values via min-max scaling. Following this, KMeans clustering is applied on $F^{simi}$ to construct $N$ similarity-based clusters $C = \{C_1, ..., C_N\}$, ensuring the categorization adheres to both the neighborhood similarity and confidence conditions.

Once the categorization is completed, the original predictions for nodes within each designated group $C_n$ are scaled by a *group-specific* temperature $\boldsymbol{T}_n$, tailored to address nearby affinity status of the

Table 2: ECE results (reported in percentage) for our proposed calibration method and baselines. A lower ECE indicates better calibration performance. The best and second best performances are represented by bold and underline texts, respectively.

| Methods | | UnCal. | TS | VS | ETS | CaGCN | GATS | Ours |
|---|---|---|---|---|---|---|---|---|
| Cora | GCN | $12.43 \pm 4.24$ | $3.87 \pm 1.22$ | $4.30 \pm 1.28$ | $3.78 \pm 1.25$ | $5.22 \pm 1.45$ | $\underline{3.55 \pm 1.28}$ | $\mathbf{1.97 \pm 0.44}$ |
| | GAT | $14.88 \pm 4.30$ | $3.42 \pm 1.00$ | $3.45 \pm 1.13$ | $3.32 \pm 0.92$ | $3.81 \pm 1.00$ | $\underline{3.05 \pm 0.78}$ | $\mathbf{2.08 \pm 0.45}$ |
| Citeseer | GCN | $12.54 \pm 8.58$ | $5.27 \pm 1.70$ | $5.15 \pm 1.46$ | $5.10 \pm 1.76$ | $6.60 \pm 1.76$ | $\underline{4.49 \pm 1.53}$ | $\mathbf{2.66 \pm 0.53}$ |
| | GAT | $16.65 \pm 7.98$ | $5.08 \pm 1.48$ | $4.62 \pm 1.58$ | $5.01 \pm 1.46$ | $4.86 \pm 1.68$ | $\underline{4.01 \pm 1.42}$ | $\mathbf{2.86 \pm 0.56}$ |
| Pubmed | GCN | $7.30 \pm 1.56$ | $1.27 \pm 0.30$ | $1.46 \pm 0.29$ | $1.26 \pm 0.31$ | $1.05 \pm 0.33$ | $\underline{0.95 \pm 0.32}$ | $\mathbf{0.75 \pm 0.15}$ |
| | GAT | $10.38 \pm 1.89$ | $1.15 \pm 0.46$ | $1.05 \pm 0.36$ | $1.13 \pm 0.47$ | $0.99 \pm 0.34$ | $\underline{0.98 \pm 0.36}$ | $\mathbf{0.69 \pm 0.16}$ |
| Computers | GCN | $2.96 \pm 0.76$ | $2.62 \pm 0.55$ | $2.70 \pm 0.61$ | $2.59 \pm 0.72$ | $\underline{1.70 \pm 0.53}$ | $2.15 \pm 0.52$ | $\mathbf{1.02 \pm 0.26}$ |
| | GAT | $1.58 \pm 0.56$ | $1.44 \pm 0.35$ | $1.44 \pm 0.40$ | $1.42 \pm 0.43$ | $1.82 \pm 0.63$ | $\underline{1.36 \pm 0.34}$ | $\mathbf{0.95 \pm 0.37}$ |
| Photo | GCN | $2.11 \pm 0.97$ | $1.68 \pm 0.68$ | $1.75 \pm 0.67$ | $1.63 \pm 0.84$ | $1.98 \pm 0.53$ | $\underline{1.46 \pm 0.51}$ | $\mathbf{1.01 \pm 0.36}$ |
| | GAT | $2.18 \pm 1.54$ | $1.56 \pm 0.63$ | $1.65 \pm 0.70$ | $1.57 \pm 0.78$ | $2.04 \pm 0.74$ | $\underline{1.49 \pm 0.65}$ | $\mathbf{0.97 \pm 0.53}$ |
| CS | GCN | $1.72 \pm 1.28$ | $1.01 \pm 0.24$ | $0.94 \pm 0.28$ | $0.97 \pm 0.22$ | $2.32 \pm 1.12$ | $\underline{0.90 \pm 0.29}$ | $\mathbf{0.58 \pm 0.19}$ |
| | GAT | $1.48 \pm 0.79$ | $1.07 \pm 0.34$ | $1.01 \pm 0.40$ | $1.03 \pm 0.31$ | $2.27 \pm 1.13$ | $\underline{0.85 \pm 0.23}$ | $\mathbf{0.72 \pm 0.43}$ |
| Physics | GCN | $0.56 \pm 0.33$ | $0.51 \pm 0.19$ | $0.46 \pm 0.15$ | $0.51 \pm 0.19$ | $0.88 \pm 0.47$ | $\underline{0.45 \pm 0.15}$ | $\mathbf{0.28 \pm 0.11}$ |
| | GAT | $0.55 \pm 0.24$ | $0.56 \pm 0.20$ | $0.56 \pm 0.21$ | $0.55 \pm 0.20$ | $1.06 \pm 0.40$ | $\mathbf{0.43 \pm 0.16}$ | $\underline{0.48 \pm 0.22}$ |
| CoraFull | GCN | $6.49 \pm 1.28$ | $5.55 \pm 0.45$ | $5.79 \pm 0.43$ | $5.49 \pm 0.46$ | $5.92 \pm 2.84$ | $\underline{3.74 \pm 0.63}$ | $\mathbf{3.46 \pm 1.31}$ |
| | GAT | $5.25 \pm 1.32$ | $4.41 \pm 0.50$ | $4.42 \pm 0.49$ | $4.36 \pm 0.50$ | $6.80 \pm 3.81$ | $\underline{3.46 \pm 0.46}$ | $\mathbf{2.64 \pm 1.02}$ |

Table 3: ECE results (in percentage) for our method and baselines on large-scale datasets, with lower ECE indicates better performance.

| Methods | | UnCal. | CaGCN | GATS | Ours |
|---|---|---|---|---|---|
| Arxiv | GCN | $4.92 \pm 0.36$ | $1.97 \pm 0.16$ | $0.75 \pm 0.06$ | $\mathbf{0.71 \pm 0.13}$ |
| | SAGE | $3.00 \pm 0.89$ | $1.84 \pm 0.19$ | $2.05 \pm 0.28$ | $\mathbf{0.98 \pm 0.23}$ |
| Reddit | GCN | $8.55 \pm 1.28$ | $1.86 \pm 0.19$ | $2.56 \pm 0.59$ | $\mathbf{0.35 \pm 0.05}$ |
| | SAGE | $11.30 \pm 1.99$ | $2.14 \pm 0.35$ | $4.66 \pm 0.57$ | $\mathbf{0.73 \pm 0.15}$ |

Table 4: Calibration duration (in seconds) for our method and baselines on large-scale datasets, with lower values denoting improved efficiency.

| Methods | | CaGCN | GATS | Ours |
|---|---|---|---|---|
| Arxiv | GCN | $20.84 \pm 2.69$ | $48.89 \pm 11.39$ | $\mathbf{7.10 \pm 0.94}$ **(-41.79 sec)** |
| | SAGE | $23.02 \pm 4.44$ | $61.67 \pm 16.89$ | $\mathbf{4.85 \pm 0.65}$ **(-56.82 sec)** |
| Reddit | GCN | $55.98 \pm 13.76$ | $72.90 \pm 19.98$ | $\mathbf{11.04 \pm 0.30}$ **(-61.86 sec)** |
| | SAGE | $78.13 \pm 27.35$ | $192.01 \pm 177.57$ | $\mathbf{9.91 \pm 0.95}$ **(-182.1 sec)** |

$n$-th cluster:

$$\widetilde{p}_i = \max_k \sigma_{\mathrm{sm}} \left( \frac{z_i}{\boldsymbol{T}_n} \right)_k \in \mathbb{R}, \quad i \in C_n. \tag{9}$$

The array of group-wise temperatures $\boldsymbol{T} \in \mathbb{R}^N$ is then optimized through the standard cross-entropy loss $\mathcal{L}_{\mathrm{CE}}$ and an auxiliary regularization loss $\mathcal{L}_{simi}$. Given the availability of a validation set during the post-hoc calibration phase, $\mathcal{L}_{simi}$ is integrated to minimize the discrepancy between the average scaled confidence $\widetilde{p}_i$ for all nodes and the accuracy of validation nodes, represented as $a_{val}^{(m)}$, within distinct groups:

$$\mathcal{L} = \mathcal{L}_{\mathrm{CE}} + \lambda \mathcal{L}_{simi},$$

$$\mathcal{L}_{simi} = \sum_{n=1}^{N} ||a_{val}^{(n)} - \frac{1}{|C_n|} \sum_{i \in C_n} \widetilde{p}_i||^2, \tag{10}$$

where $\lambda$ serves as a scaling factor for $\mathcal{L}_{simi}$. Throughout this procedure, our method effectively rectifies miscalibrations across varying neighborhood topologies by organizing nodes with analogous local similarity levels and applying per-cluster temperatures, specialized to refine the miscalibration within each designated group. The overall pipeline of SIMI-MAILBOX is illustrated in Figure 2.

## 6 EXPERIMENTS

In our studies, we validate the efficacy of the proposed method under extensive experiments, leveraging two representative GNN architectures: GCN and GAT. We evaluate the performance of our

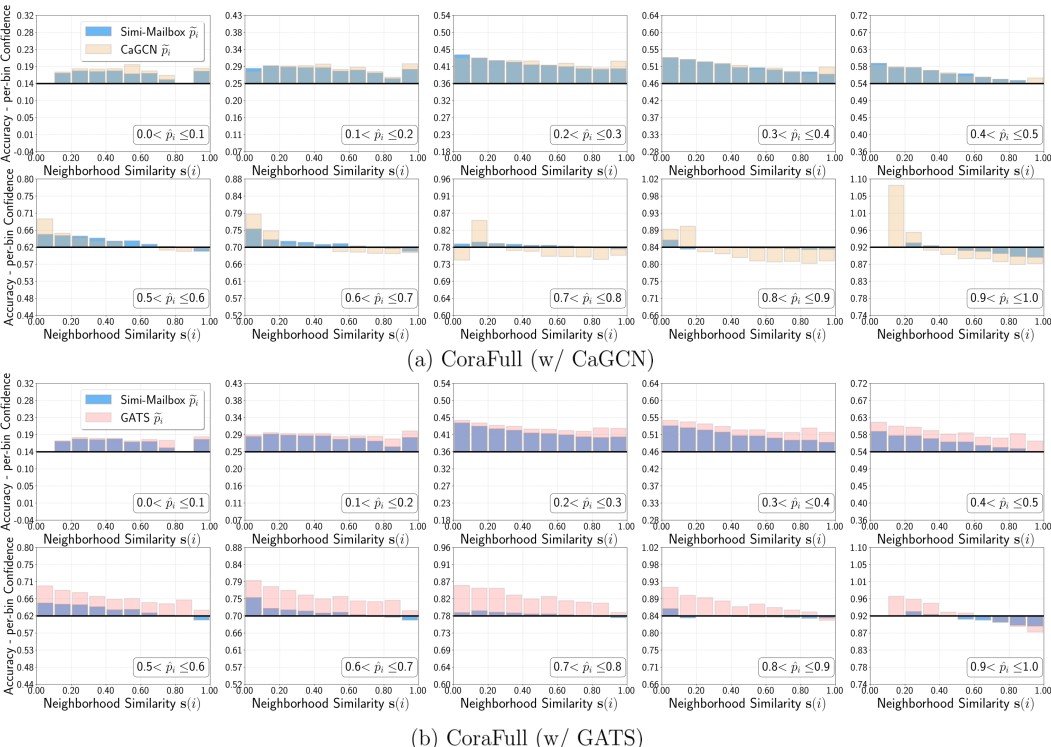

(a) CoraFull (w/ CaGCN)

(b) CoraFull (w/ GATS)

Figure 3: Qualitative analysis of our calibration results on CoraFull dataset, compared with CaGCN and GATS. The accuracy in per-confidence interval is represented as **black** horizontal lines. The **blue**, **yellow** and **pink** bars illustrate the discrepancy between per-bin accuracy and average confidence of nodes within each affinity sub-interval, calibrated through our SIMI-MAILBOX, CaGCN, and GATS, respectively. Throughout the confidence intervals, our method facilitates a better reduction in the gap between accuracy and confidence across diverse affinity levels, compared to baseline methods.

SIMI-MAILBOX across eight benchmark datasets including citation datasets: Cora (Sen et al., 2008), Citeseer (Sen et al., 2008), Pubmed (Sen et al., 2008), CoraFull (Bojchevski & Günnemann, 2017), Coauthor CS (Shchur et al., 2018), and Coauthor Physics (Shchur et al., 2018), alongside Amazon co-purchase datasets: Computers (Shchur et al., 2018) and Photo (Shchur et al., 2018). To further demonstrate the versatility, we extended our validation to large-scale datasets, Arxiv (Hu et al., 2020) and Reddit (Zeng et al., 2019). Detailed dataset statistics can be found in Appendix A.

## 6.1 PERFORMANCE EVALUATION

**Experimental Setup.** We undertake our experiments following the experimental protocols of GATS (Hsu et al., 2022b) in the scope of semi-supervised node classification. Comprehensive details of the experiment configurations are provided in Appendix A. In the post-hoc calibration phase, the validation set is employed to train calibration models. Thus, the optimal calibration models are selected based on the lowest validation ECE on the training set. To assess the calibration performance, we use ECE as a metric (Naeini et al., 2015), following the common practice (Wang et al., 2021b; Hsu et al., 2022b). Additional calibration metrics, including class-wise ECE (Kull et al., 2019; Nixon et al., 2019), Kernel Density Estimation-based ECE (KDE-ECE) (Zhang et al., 2020), Brier Score (Brier et al., 1950), and Negative Log-likelihood (NLL), are provided in Appendix B.

**Baselines.** In alignment with precedent studies, we compare our method against classical calibration methods: Temperature scaling (TS) (Guo et al., 2017), Vector scaling (VS) (Guo et al., 2017), and Ensemble temperature sclaing (ETS) (Zhang et al., 2020) and GNN-specialized calibration baselines: GCN as a calibration function (CaGCN) (Wang et al., 2021b) and Graph attention temperature scaling (GATS) (Hsu et al., 2022b).

**Results.** Table 2 presents the calibration results of our SIMI-MAILBOX in contrast with baseline methods measured using the ECE metric. Overall, our method achieves state-of-the-art performance across 15 out of 16 scenarios, displaying a substantial performance divergence compared to the baselines. More precisely, our method pioneers in achieving an error rate below $3\%$ on Cora and Citeseer datasets, demonstrating marked predominance on Cora with GCN by first breaking into the $1\%$ error range. This superiority is also witnessed in both Pubmed and CS, decreasing ECE within the $[0.5, 0.8]$ range for the first time. Moreover, SIMI-MAILBOX surpasses existing methods on Amazon datasets (Computers and Photo), reducing calibration errors to below $1\%$ with GAT. Significant performance advancements are evident in Physics and CoraFull as well, first achieving error rates in the $0.2\%$ and $2.64\%$ range, respectively. This demonstrates the effectiveness of our method in rectifying the miscalibration of conventional GNNs across diverse benchmark datasets and architectures.

## 6.2 SIMI-MAILBOX ON LARGE-SCALE GRAPHS

Beyond the assessment in regular-scale datasets, we extend our experiments to large-scale graphs to emphasize the versatility of our method in handling graphs of varying sizes. In this experiment, we adopt GCN and GraphSAGE (SAGE) (Hamilton et al., 2017), since they are representative architectures for our large-scale benchmark datasets. We adhere to the same evaluation protocol in Hu et al. (2020) to compare the performance of SIMI-MAILBOX with other GNN-based calibration methods, CaGCN and GATS. As evident in Table 3, SIMI-MAILBOX outperforms all baselines to a considerable extent, accomplishing an error rate below $1\%$ in all examined settings. This superiority is particularly observed in Reddit with SAGE, where our method refines miscalibration by 10.57 compared to the uncalibrated outcome. Besides evaluating calibration performance, we also quantified the total execution duration in seconds for each run, presented in Table 4. According to the table, our method significantly enhances time efficiency across all experiments, with a substantial reduction observed in the Reddit dataset, decreasing by 61.86 and 182.10 seconds for GCN and SAGE, correspondingly.

## 6.3 EFFECTIVENESS ON DIVERSE NEIGHBORHOOD TOPOLOGY

To further substantiate the effectiveness of our SIMI-MAILBOX on refining miscalibrations across diverse neighborhood similarity levels, we provide a qualitative comparison in Figure 3, utilizing a consistent dataset (CoraFull) and GNN architecture (GCN) in Section 4. Analogous to the previous section, we represent per-confidence bin accuracy as **black** horizontal lines in each figure. In this analysis, the **blue** bars illustrate the difference between the accuracy and average confidence of calibrated nodes within each neighborhood affinity sub-interval, refined by our method. Conversely, the deviations concerning CaGCN and GATS are represented by **yellow** and **pink** bars, respectively. As illustrated, our method facilitates a more pronounced reduction in the gap between accuracy and average confidence across varying affinity, compared to baseline methods. This superiority is clearly pronounced for patterns mentioned in Section 4, mitigating a significant discrepancy in $s(i) \in (0.1, 0.2]$ within $\hat{p}_i \in (0.9, 1.0]$ interval on CaGCN and a prevailing underconfidence concerning $s(i) \in (0., 0.4]$ in the $\hat{p}_i \in (0.2, 0.4]$ and $\hat{p}_i \in (0.6, 0.8]$ confidence areas.

## 7 CONCLUSION

In this study, we unveiled a novel analysis highlighting the inconsistency between neighborhood affinity and foundational design philosophies of existing calibration research. In response to the identified limitations, we proposed SIMI-MAILBOX, a novel calibration method that categorizes and group-speficically refines nodes, considering both neighborhood similarity and inherent confidence, leveraging our crucial insight that nodes with analogous local topology and confidence share similar calibration errors. While extensive validations have substantiated SIMI-MAILBOX's efficacy, the scope of our approach primarily concentrates on homophilous graphs. As a direction for future work, it would be desirable to explore the calibration status of GNNs specialized in heterophilous graphs.

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

# SUPPLEMENTARY MATERIALS

## A   DETAILED EXPERIMENTAL SETUP

### A.1   DATASET STATISTICS

Table 5 provides comprehensive statistics of the datasets used in our experiments, including the number of nodes, edges, classes, and features.

Table 5: Statistics of benchmark datasets.

| Dataset | #Nodes | #Edges | #Classes | #Features |
|---------|--------|--------|----------|-----------|
| Cora | 2,708 | 10,556 | 7 | 1,433 |
| Citeseer | 3,327 | 9,104 | 6 | 3,703 |
| Pubmed | 19,717 | 88,648 | 3 | 500 |
| CoraFull | 19,793 | 126,842 | 70 | 8,710 |
| Computers | 13,752 | 491,722 | 10 | 767 |
| Photo | 7,650 | 238,162 | 8 | 745 |
| CS | 18,333 | 163,788 | 15 | 6,805 |
| Physics | 34,493 | 495,924 | 5 | 8,415 |
| Arxiv | 169,343 | 1,166,243 | 40 | 128 |
| Reddit | 232,965 | 23,213,838 | 41 | 602 |

### A.2   IMPLEMENTATION DETAILS

We implement GNN models and the proposed method using PyTorch (Paszke et al., 2019) and PyTorch Geometric (Fey & Lenssen, 2019). The experiments are conducted on RTX 2080ti (11GB) and RTX 3090ti GPU (24G). The experimental settings and evaluation protocols are largely consistent with those used in the GATS framework (Hsu et al., 2022b). We split the labeled and unlabeled data by 15% and 85%, respectively, and perform three-fold cross-validation on the former, 10% for training and the remaining 5% for validation. We conduct 75 runs in total for each experiment, considering 5 random splits, 5 random initializations, and three-fold cross-validation. For optimization, we select the Adam optimizer with an initial learning rate of 0.01. The weight decay values are set to 5e-4 for Cora, Citeseeer, and Pubmed and 0 for the remaining datasets. The GNN architectures are configured as follows: for GCN (Kipf & Welling, 2016), we employ two GNN layers with 64 hidden units; for GAT (Veličković et al., 2017), the number of attention heads is set to 8, with 8 hidden units per head. We train GNNs in a maximum of 2000 epochs with early stopping based on a patience of 100 epochs. A dropout rate of 0.5 is applied uniformly across all backbones. To ensure fair comparisons, we refer to the implementation and setup details provided in the released code of GATS (Hsu et al., 2022b) for the baseline methods. For experiments on large-scale datasets with GCN and SAGE (Hamilton et al., 2017), we adhere to the same split ratio and evaluation protocol in (Hu et al., 2020), and report ECE results, averaged over ten random seeds. We employ three-layer and two-layer GNN on Arxiv and Reddit, respectivly, while fixing the hidden dimension as 256 across both GCN and SAGE. Analogous to the regular-scale experiment, we select the Adam optimizer with an initial learning rate of 0.01 and weight decay as 0 uniformly across all large-scale settings. Meanwhile, we construct hyperparameter search spaces for proposed SIMI-MAILBOX as follows. The number of clusters $N$ in our method is selected from the range $[5, 30]$. The regularization coefficient $\lambda$ is explored in the range of $[1, 50]$ During the evaluation, we set the number of bins for ECE and class-wise ECE measurements as 15, following prior works. The source code for our experiments is available at https://anonymous.4open.science/r/Simi_Mailbox-0816/

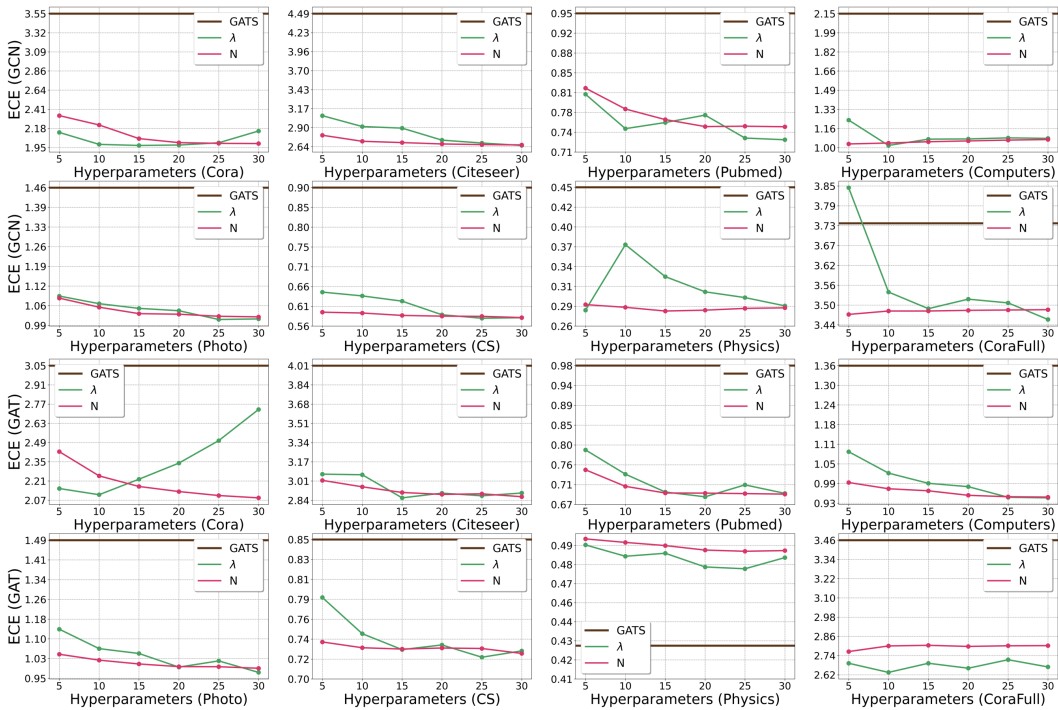

Figure 4: Hyperparameter sensitivity of scaling factor $\lambda$ and the number of bins $N$ across all benchmark datasets and GNN architectures.

Table 6: ECE results (reported in percentage) for our original calibration method with min-max scaling (specified as **Ours**) and standard scaling (specified as **Ours w/ standard scaling**), compared to GATS. A lower ECE indicates better calibration performance. The best and second best performances are represented by bold and underline texts, respectively.

| | Datasets | Cora | Citeseer | Pubmed | Computers | Photo | CS | Physics | CoraFull |
|---|---|---|---|---|---|---|---|---|---|
| | GATS | $3.55_{\pm 1.28}$ | $4.49_{\pm 1.53}$ | $0.95_{\pm 0.32}$ | $2.15_{\pm 0.52}$ | $1.46_{\pm 0.51}$ | $0.90_{\pm 0.29}$ | $0.45_{\pm 0.15}$ | $3.74_{\pm 0.63}$ |
| GCN | **Ours** | $\mathbf{1.97}_{\pm 0.44}$ | $\mathbf{2.66}_{\pm 0.53}$ | $\mathbf{0.75}_{\pm 0.15}$ | $\mathbf{1.02}_{\pm 0.26}$ | $\mathbf{1.01}_{\pm 0.36}$ | $\mathbf{0.58}_{\pm 0.19}$ | $\mathbf{0.28}_{\pm 0.11}$ | $\mathbf{3.46}_{\pm 1.31}$ |
| | **Ours w/ standard scaling** | $\underline{2.02}_{\pm 0.54}$ | $\underline{2.67}_{\pm 0.52}$ | $\mathbf{0.75}_{\pm 0.14}$ | $\underline{1.06}_{\pm 0.24}$ | $\mathbf{1.01}_{\pm 0.34}$ | $\underline{0.63}_{\pm 0.18}$ | $\underline{0.30}_{\pm 0.10}$ | $\underline{3.53}_{\pm 1.30}$ |
| | GATS | $3.05_{\pm 0.78}$ | $4.01_{\pm 1.42}$ | $0.98_{\pm 0.36}$ | $1.36_{\pm 0.34}$ | $1.49_{\pm 0.65}$ | $0.85_{\pm 0.23}$ | $\mathbf{0.43}_{\pm 0.16}$ | $3.46_{\pm 0.46}$ |
| GAT | **Ours** | $\mathbf{2.08}_{\pm 0.45}$ | $\mathbf{2.86}_{\pm 0.56}$ | $\mathbf{0.69}_{\pm 0.16}$ | $\underline{0.95}_{\pm 0.37}$ | $\mathbf{0.97}_{\pm 0.53}$ | $\mathbf{0.72}_{\pm 0.43}$ | $\underline{0.48}_{\pm 0.22}$ | $\mathbf{2.64}_{\pm 1.02}$ |
| | **Ours w/ standard scaling** | $\underline{2.23}_{\pm 0.41}$ | $\underline{2.93}_{\pm 0.58}$ | $\mathbf{0.69}_{\pm 0.19}$ | $\mathbf{0.94}_{\pm 0.37}$ | $\underline{0.98}_{\pm 0.52}$ | $\underline{0.73}_{\pm 0.43}$ | $0.49_{\pm 0.21}$ | $\underline{2.82}_{\pm 1.06}$ |

# B ADDITIONAL DISCUSSIONS AND EXPERIMENTAL RESULTS

## B.1 ALGORITHMIC POINT OF VIEW ON PREVIOUS GNN CALIBRATION STUDIES

This subsection provides the limitation of previous GNN calibration approaches for addressing varying similarity levels in the algorithmic perspective. To begin with, the node-wise temperature $\boldsymbol{T}^{\text{CaGCN}}$ for $l$ layers in CaGCN (Wang et al., 2021b) is defined as below:

$$\boldsymbol{T}^{\text{CaGCN}} = \sigma^+(\boldsymbol{A}\sigma_{\text{ReLU}}(...\boldsymbol{A}\sigma_{\text{ReLU}}(\boldsymbol{A}\boldsymbol{Z}W^{(1)})W^{(2)}...)W^{(l)}) \in \mathbb{R}^{|\mathcal{V}|}, \qquad (11)$$

where $\sigma^+$ and $\sigma_{\text{ReLU}}$ denote softplus and ReLU operation, while $\boldsymbol{Z}$ and $W$ represent logits from trained GNNs and trainable weights, respectively.

The foundational assumption of CaGCN for leveraging GCN as a temperature function is that confidence for nodes linked to agreeing nodes should elevate, while that for nodes with disagreeing neighbors should decrease. They assert that GCN can make the confidence of adjacent nodes similar by propagating the predictions to neighboring counterparts. However, our findings in Section 4 indicate that this does not hold true, especially for nodes with dissimilar neighbors. Moreover,

Table 7: ECE results (reported in percentage) on heterophilous datasets for our proposed calibration method and baselines, averaged over 50 runs. A lower ECE indicates better calibration performance.

| Datasets | | Chameleon | Squirrel | Actor | Texas | Wisconsin | Cornell |
|---|---|---|---|---|---|---|---|
| GCN | UnCal. | $9.39 \pm 1.90$ | $7.23 \pm 1.46$ | $\mathbf{2.69} \pm \mathbf{0.88}$ | $18.15 \pm 4.13$ | $17.76 \pm 6.97$ | $19.17 \pm 4.94$ |
| | TS | $9.42 \pm 1.94$ | $7.18 \pm 1.37$ | $2.82 \pm 0.95$ | $18.12 \pm 4.95$ | $15.41 \pm 5.08$ | $19.93 \pm 5.31$ |
| | GATS | $8.25 \pm 2.15$ | $6.85 \pm 1.09$ | $2.82 \pm 1.01$ | $18.73 \pm 4.34$ | $15.76 \pm 5.33$ | $21.60 \pm 5.28$ |
| | Ours | $\mathbf{7.50} \pm \mathbf{1.40}$ | $\mathbf{5.40} \pm \mathbf{1.04}$ | $2.73 \pm 0.89$ | $\mathbf{15.50} \pm \mathbf{4.44}$ | $\mathbf{15.19} \pm \mathbf{3.58}$ | $\mathbf{18.66} \pm \mathbf{5.24}$ |
| GAT | UnCal. | $7.27 \pm 1.43$ | $6.42 \pm 1.30$ | $3.49 \pm 1.11$ | $18.51 \pm 4.47$ | $16.12 \pm 4.36$ | $\mathbf{14.89} \pm \mathbf{6.35}$ |
| | TS | $7.23 \pm 1.44$ | $6.43 \pm 1.31$ | $3.29 \pm 1.15$ | $18.62 \pm 3.99$ | $15.64 \pm 3.25$ | $16.00 \pm 6.72$ |
| | GATS | $7.79 \pm 1.95$ | $6.66 \pm 1.63$ | $3.41 \pm 1.14$ | $18.91 \pm 4.49$ | $15.16 \pm 2.86$ | $18.08 \pm 6.74$ |
| | Ours | $\mathbf{6.75} \pm \mathbf{1.73}$ | $\mathbf{5.45} \pm \mathbf{1.30}$ | $\mathbf{2.64} \pm \mathbf{0.99}$ | $\mathbf{15.53} \pm \mathbf{3.48}$ | $\mathbf{14.90} \pm \mathbf{3.29}$ | $18.54 \pm 6.35$ |

according to the above formulation, the temperature function **does not necessarily yield** higher temperatures for nodes with dissimilar neighbors or lower ones for those with similar neighbors, leading to suboptimal calibration results across diverse neighborhood affinity level.

Meanwhile, from the perspective of individual nodes $i$, the temperature function of GATS (Hsu et al., 2022b) $\boldsymbol{T}_i^{\text{GATS}}$ is formulated as:

$$\boldsymbol{T}_i^{\text{GATS}} = \frac{1}{H} \sum_{h=1}^{H} \sigma^+ (\omega \, \delta \hat{c}_i + \sum_{j \in \mathcal{N}_i} \alpha_{ij} \gamma_j \tau_j^h) + T_0 \in \mathbb{R}. \quad (12)$$

Here, $H$ and $T_0$ signify the number of attention heads and the initial bias term, respectively, with $\omega$ acting as a learnable coefficient to scale the relative confidence $\delta \hat{c}_i$ against neighborhood. The scaling factor $\gamma$ is introduced to leverage the distance-to-training-nodes property, and $\tau_j$ refers to the original logits $z_j$ transformed by a linear layer, followed by class-wise sorting within individual nodes' logits.

Recall that GATS demonstrated an increment in calibration error with a decrement in representational similarity, thereby introducing attention coefficient $\alpha_{ij}$ to reflect this. While $\alpha_{ij}$ attempts to capture the affinity between nodes, the model's capacity to discern and appropriately adjust for low similarity levels is limited, since the **complex integration of various factors** may lead to suboptimal temperature adjustments. For instance, the impact of the initial bias term $T_0$ and $\omega \delta \hat{c}_i$ may obscure the neighborhood similarity associated with nodes, which may not adequately capture the distinct calibration needs of nodes in diverse similarity contexts. Consequently, nodes in low or high similarity contexts might receive suboptimal temperature adjustments.

## B.2 HYPERPARAMETER SENSITIVITY

We present the comprehensive sensitivity analysis on robustness of SIMI-MAILBOX with respect to its hyperparameters and the choice of scaling functions. We compare the ECE results of the strongest baseline GATS (specified as dark brown) and ours across varying values of a scaling factor $\lambda$ (specified as green) and the number of bins $N$ (specified as pink) within the range of [5, 10, 15, 20, 25, 30]. The results on both GCN and GAT across whole benchmark datasets is depicted in Figure 4.

As demonstrated in the figure, SIMI-MAILBOX consistently outperforms the baseline across all hyperparameter configurations throughout diverse settings, with the number of bins $N$ demonstrating particularly stable performance trends. This robustness is attributed to our method's design, which accounts for the correlation between neighborhood similarity and the degree of miscalibration.

Meanwhile, the choice of min-max scaling is rooted in the intuition of potential disparity in distributions of neighborhood similarity and confidence. For instance, while neighborhood similarity can be evenly distributed between 0 and 1, confidence in a high-accuracy dataset might be concentrated at higher values. In this situation, min-max scaling is an effective technique for normalizing data, especially when the values are concentrated in a specific range.

Table 8: KDE-ECE results (reported in percentage) for our proposed calibration method and baselines, averaged over 75 repetitions ($\pm$ STD). A lower value indicates better calibration performance.

| Methods | | UnCal. | TS | VS | ETS | CaGCN | GATS | Ours |
|---|---|---|---|---|---|---|---|---|
| Cora | GCN | $12.76 \pm 4.09$ | $3.13 \pm 1.11$ | $3.52 \pm 1.17$ | $3.16 \pm 1.11$ | $4.01 \pm 1.40$ | $2.95 \pm 1.23$ | $\mathbf{1.88} \pm \mathbf{0.26}$ |
| | GAT | $15.02 \pm 4.23$ | $2.85 \pm 0.80$ | $2.84 \pm 0.85$ | $2.86 \pm 0.76$ | $3.38 \pm 1.29$ | $2.64 \pm 0.77$ | $\mathbf{1.98} \pm \mathbf{0.42}$ |
| Citeseer | GCN | $12.59 \pm 8.53$ | $4.90 \pm 1.67$ | $4.79 \pm 1.47$ | $4.74 \pm 1.72$ | $6.07 \pm 1.76$ | $4.25 \pm 1.49$ | $\mathbf{2.40} \pm \mathbf{0.43}$ |
| | GAT | $16.64 \pm 7.96$ | $4.74 \pm 1.42$ | $4.29 \pm 1.15$ | $4.67 \pm 1.40$ | $4.57 \pm 1.79$ | $3.84 \pm 1.45$ | $\mathbf{2.66} \pm \mathbf{0.50}$ |
| Pubmed | GCN | $7.44 \pm 1.53$ | $1.33 \pm 0.28$ | $1.58 \pm 0.38$ | $1.38 \pm 0.29$ | $1.25 \pm 0.35$ | $1.06 \pm 0.26$ | $\mathbf{0.93} \pm \mathbf{0.12}$ |
| | GAT | $10.38 \pm 1.88$ | $1.18 \pm 0.35$ | $1.13 \pm 0.31$ | $1.18 \pm 0.35$ | $1.08 \pm 0.29$ | $1.11 \pm 0.34$ | $\mathbf{0.79} \pm \mathbf{0.12}$ |
| Computers | GCN | $3.01 \pm 0.91$ | $2.60 \pm 0.65$ | $2.72 \pm 0.74$ | $2.73 \pm 0.76$ | $1.58 \pm 0.45$ | $2.12 \pm 0.60$ | $\mathbf{1.27} \pm \mathbf{0.15}$ |
| | GAT | $1.70 \pm 0.63$ | $1.56 \pm 0.42$ | $1.59 \pm 0.46$ | $1.63 \pm 0.47$ | $1.64 \pm 0.46$ | $1.52 \pm 0.40$ | $\mathbf{1.10} \pm \mathbf{0.20}$ |
| Photo | GCN | $2.45 \pm 1.23$ | $1.81 \pm 0.92$ | $1.95 \pm 0.95$ | $1.88 \pm 0.98$ | $1.64 \pm 0.42$ | $1.65 \pm 0.68$ | $\mathbf{1.13} \pm \mathbf{0.18}$ |
| | GAT | $2.42 \pm 1.60$ | $1.70 \pm 0.73$ | $1.79 \pm 0.78$ | $1.77 \pm 0.80$ | $1.73 \pm 0.62$ | $1.72 \pm 0.70$ | $\mathbf{1.19} \pm \mathbf{0.29}$ |
| CS | GCN | $2.19 \pm 1.33$ | $1.12 \pm 0.10$ | $1.12 \pm 0.18$ | $1.12 \pm 0.10$ | $1.94 \pm 0.90$ | $1.08 \pm 0.12$ | $\mathbf{0.94} \pm \mathbf{0.11}$ |
| | GAT | $1.77 \pm 0.91$ | $1.12 \pm 0.23$ | $1.12 \pm 0.25$ | $1.13 \pm 0.23$ | $1.90 \pm 0.97$ | $1.13 \pm 0.20$ | $\mathbf{0.95} \pm \mathbf{0.22}$ |
| Physics | GCN | $0.97 \pm 0.31$ | $0.83 \pm 0.09$ | $0.82 \pm 0.07$ | $0.83 \pm 0.09$ | $0.93 \pm 0.19$ | $0.85 \pm 0.09$ | $\mathbf{0.70} \pm \mathbf{0.61}$ |
| | GAT | $0.86 \pm 0.15$ | $0.84 \pm 0.10$ | $0.86 \pm 0.09$ | $0.84 \pm 0.10$ | $1.03 \pm 0.21$ | $0.82 \pm 0.08$ | $\mathbf{0.80} \pm \mathbf{0.11}$ |
| CoraFull | GCN | $6.44 \pm 1.33$ | $5.46 \pm 0.44$ | $5.68 \pm 0.41$ | $5.42 \pm 0.46$ | $5.74 \pm 0.46$ | $3.70 \pm 0.65$ | $\mathbf{3.43} \pm \mathbf{1.27}$ |
| | GAT | $5.26 \pm 1.38$ | $4.34 \pm 0.48$ | $4.36 \pm 0.46$ | $4.30 \pm 0.48$ | $6.59 \pm 3.62$ | $3.46 \pm 0.45$ | $\mathbf{2.64} \pm \mathbf{0.98}$ |

However, our method can accomplish prominent performance when equipped with different scaling functions. To verify this, we conducted additional experiment on our method with standard scaling (standard normalization) for constructing a feature vector, illustrated in Table 6. According to the table, SIMI-MAILBOX equipped with standard scaling consistently outperforms the strongest baseline in 15 of the 16 settings. Moreover, the performance gap between our method with standard scaling and the original SIMI-MAILBOX is marginal, suggesting that SIMI-MAILBOX is resilient to different choices of scaling method as well.

## B.3 RESULTS ON HETEROPHILOUS GRAPHS

We conducted additional evaluations to to further demonstrate the efficacy of our SIMI-MAILBOX on heterophilous graphs, in comparison with uncalibrated GNNs (UnCal), temperature scaling (TS) (Guo et al., 2017), and GATS. Our benchmark datasets for this experiment included Chameleon, Squirrel, Actor, Texas, Wisconsin, and Cornell (Rozemberczki et al., 2021; Pei et al., 2020). We adopted 10 different train/validation/test splits provided in the official PyTorch Geometric Library (Fey & Lenssen, 2019). For each split, we conducted 5 random initialization, resulting in 50 runs in total. We maintained the same seeds to our method and the baseline.

As indicated in Table 7, our method surpasses the baselines in 14 out of 16 settings. Notably, on the Texas dataset , SIMI-MAILBOX achieves ECE reduction of 2.65% and 2.98% for GCN and GAT, respectively, compared to uncalibrated results. Conversely, TS and GATS showed limited effectiveness in reducing calibration error in the Texas; GATS in fact increases ECE beyond the uncalibrated results. While the improvements with heterophilous graphs are less pronounced than those observed with homophilous graphs, SIMI-MAILBOX still effectively mitigates miscalibration against previous calibration methods. This is attributed to our method's careful categorization on the basis of neighborhood similarity and confidence levels.

## B.4 RESULTS FROM DIFFERENT EVALUATION METRICS

We here provide supplementary results evaluated with different calibration metrics, including class-wise ECE (Kull et al., 2019), kernel density estimation-based ECE (KDE-ECE) (Zhang et al., 2020), negative log-likelihood (NLL), and Brier Score (Brier et al., 1950).

Table 9: Class-wise ECE results (reported in percentage) for our proposed calibration method and baselines, averaged over 75 repetitions ($\pm$ STD). A lower value indicates better calibration performance.

| Methods | | UnCal. | TS | VS | ETS | CaGCN | GATS | **Ours** |
|---|---|---|---|---|---|---|---|---|
| Cora | GCN | 4.14 ± 1.10 | 2.03 ± 0.23 | 2.09 ± 0.27 | 2.03 ± 0.23 | 2.21 ± 0.28 | 1.99 ± 0.24 | **1.82 ± 0.19** |
| | GAT | 4.78 ± 1.18 | 1.95 ± 0.23 | 1.94 ± 0.25 | 1.94 ± 0.23 | 2.10 ± 0.29 | 1.92 ± 0.24 | **1.80 ± 0.22** |
| Citeseer | GCN | 5.11 ± 2.77 | 2.97 ± 0.65 | 2.80 ± 0.43 | 2.94 ± 0.69 | 3.24 ± 0.78 | 2.88 ± 0.78 | **2.53 ± 0.55** |
| | GAT | 6.39 ± 2.52 | 3.03 ± 0.47 | 2.85 ± 0.48 | 3.02 ± 0.48 | 3.07 ± 0.69 | 2.96 ± 0.54 | **2.71 ± 0.44** |
| Pubmed | GCN | 5.04 ± 1.04 | 1.39 ± 0.28 | 1.54 ± 0.31 | 1.40 ± 0.27 | 1.33 ± 0.32 | 1.26 ± 0.28 | **1.17 ± 0.23** |
| | GAT | 7.19 ± 1.22 | 1.77 ± 0.40 | 1.75 ± 0.30 | 1.77 ± 0.40 | 1.67 ± 0.39 | 1.79 ± 0.36 | **1.63 ± 0.32** |
| Computers | GCN | 0.96 ± 0.16 | 0.92 ± 0.11 | 0.91 ± 0.13 | 0.94 ± 0.13 | 0.83 ± 0.10 | 0.88 ± 0.08 | **0.81 ± 0.08** |
| | GAT | 0.80 ± 0.13 | 0.78 ± 0.10 | 0.76 ± 0.09 | 0.80 ± 0.11 | 0.80 ± 0.10 | 0.78 ± 0.10 | **0.74 ± 0.09** |
| Photo | GCN | 0.86 ± 0.21 | 0.78 ± 0.14 | 0.81 ± 0.15 | 0.78 ± 0.16 | 0.79 ± 0.08 | 0.76 ± 0.11 | **0.67 ± 0.05** |
| | GAT | 0.96 ± 0.36 | 0.84 ± 0.17 | 0.82 ± 0.16 | 0.84 ± 0.19 | 0.86 ± 0.13 | 0.83 ± 0.18 | **0.74 ± 0.10** |
| CS | GCN | 0.40 ± 0.15 | 0.30 ± 0.03 | 0.32 ± 0.03 | **0.29 ± 0.03** | 0.42 ± 0.10 | **0.29 ± 0.03** | **0.29 ± 0.03** |
| | GAT | 0.39 ± 0.10 | 0.34 ± 0.03 | 0.34 ± 0.03 | 0.34 ± 0.03 | 0.44 ± 0.10 | 0.34 ± 0.04 | **0.33 ± 0.03** |
| Physics | GCN | 0.41 ± 0.33 | 0.36 ± 0.06 | 0.34 ± 0.04 | 0.36 ± 0.06 | 0.46 ± 0.14 | 0.36 ± 0.05 | **0.33 ± 0.04** |
| | GAT | 0.40 ± 0.08 | 0.39 ± 0.07 | **0.37 ± 0.05** | 0.39 ± 0.07 | 0.52 ± 0.13 | 0.38 ± 0.07 | 0.38 ± 0.06 |
| CoraFull | GCN | 0.35 ± 0.04 | 0.33 ± 0.02 | 0.34 ± 0.01 | 0.33 ± 0.02 | 0.34 ± 0.05 | 0.33 ± 0.02 | **0.32 ± 0.01** |
| | GAT | 0.34 ± 0.03 | 0.32 ± 0.01 | 0.32 ± 0.01 | 0.32 ± 0.01 | 0.35 ± 0.07 | **0.31 ± 0.01** | **0.31 ± 0.01** |

- **KDE-ECE** utilizes a smoothing kernel function denoted as $K_h$ with a fixed bandwidth $h$ to estimate the accuracies $\hat{\pi}$ and marginal probabilities $\hat{f}$. The calibration error is then quantified through the integration of the absolute difference between the estimated accuracy and predicted confidence $\hat{p}$, formulated as follows:

$$\text{KDE-ECE} = \int |\hat{\pi}(\hat{p}) - \hat{p}| \hat{f}(\hat{p}) d\hat{p},$$

$$\hat{\pi}(\hat{p}) = \frac{\sum_{i \in \mathcal{V}}^{N} \mathbf{1}[y_i = \hat{y}_i] \prod_{k=1}^{C} K_h(\hat{p} - p_{i,k})}{\sum_{i \in \mathcal{V}}^{N} \prod_{k=1}^{C} K_h(\hat{p} - p_{i,k})},$$ 

$$\hat{f}(\hat{p}) = \frac{h^{-1}}{N} \sum_{i \in \mathcal{V}}^{N} \prod_{k=1}^{C} K_h(\hat{p} - p_{i,k})$$

(13)

Following the precedent (Hsu et al., 2022b), we implement the Triweight Kernel function $K_h(v) = (1/h)\frac{35}{32}(1 - (v/h)^2)^3$ (de Haan, 1999), where the bandwidth is calculated as $h = 1.06\sigma N^{-1/5}$ (Scott, 2015), with $\sigma$ representing the standard deviation of the confidence here.

- **Class-wise ECE** extends the general concept of ECE to class-wise perspective. It measures the discrepancy between the ground-truth frequency and the average predicted probability within each confidence bin for each class $k$, defined as:

$$\text{ECE}(k) = \sum_{m=1}^{M} \frac{|B_m|}{N} |\text{freq}(B_{m,k}) - \text{conf}(B_{m,k})|,$$

$$\text{freq}(B_{m,k}) = \frac{1}{|B_{m,k}|} \sum_{i \in B_{m,k}} \mathbf{1}[y_i = k]$$

(14)

The overall class-wise ECE is obtained by averaging $\text{ECE}(k)$ across all classes, i.e. Class-wise ECE $= \frac{1}{C} \sum_{k=1}^{C} \text{ECE}(k)$.

- **NLL** is frequently used for evaluating calibration to assess the overall miscalibration, computed by the average of logarithms of the predicted probability for each correct class,

Table 10: NLL results (reported in percentage) for our proposed calibration method and baselines, averaged over 75 repetitions ($\pm$ STD). A lower value indicates better calibration performance.

| Methods | | UnCal. | TS | VS | ETS | CaGCN | GATS | Ours |
|---|---|---|---|---|---|---|---|---|
| Cora | GCN | $0.6199 \pm 0.0444$ | $0.5613 \pm 0.0302$ | $0.5747 \pm 0.0380$ | $0.5591 \pm 0.0291$ | $0.6622 \pm 0.0742$ | $0.5566 \pm 0.0310$ | $\mathbf{0.5429} \pm \mathbf{0.0249}$ |
| | GAT | $0.6087 \pm 0.0507$ | $0.5162 \pm 0.0238$ | $0.5228 \pm 0.0332$ | $0.5151 \pm 0.0232$ | $0.5420 \pm 0.0360$ | $0.5124 \pm 0.0209$ | $\mathbf{0.5040} \pm \mathbf{0.0201}$ |
| Citeseer | GCN | $0.9265 \pm 0.1038$ | $0.8800 \pm 0.0428$ | $0.8734 \pm 0.0234$ | $0.8770 \pm 0.0386$ | $0.9204 \pm 0.0578$ | $0.8702 \pm 0.0404$ | $\mathbf{0.8599} \pm \mathbf{0.0419}$ |
| | GAT | $0.9602 \pm 0.1025$ | $0.8762 \pm 0.0330$ | $0.8729 \pm 0.0254$ | $0.8752 \pm 0.0324$ | $0.8752 \pm 0.0291$ | $0.8715 \pm 0.0290$ | $\mathbf{0.8611} \pm \mathbf{0.0283}$ |
| Pubmed | GCN | $0.3939 \pm 0.0160$ | $0.3676 \pm 0.0072$ | $0.3679 \pm 0.0073$ | $0.3659 \pm 0.0073$ | $\mathbf{0.3582} \pm \mathbf{0.0073}$ | $0.3638 \pm 0.0069$ | $0.3627 \pm 0.0067$ |
| | GAT | $0.4382 \pm 0.0120$ | $0.3871 \pm 0.0078$ | $0.3864 \pm 0.0070$ | $0.3870 \pm 0.0078$ | $0.3845 \pm 0.0072$ | $0.3866 \pm 0.0077$ | $\mathbf{0.3844} \pm \mathbf{0.0075}$ |
| Computers | GCN | $0.4297 \pm 0.0119$ | $0.4295 \pm 0.0116$ | $0.4291 \pm 0.0113$ | $0.4130 \pm 0.0146$ | $0.4333 \pm 0.0356$ | $0.4243 \pm 0.0134$ | $\mathbf{0.4080} \pm \mathbf{0.0104}$ |
| | GAT | $0.3739 \pm 0.0145$ | $0.3734 \pm 0.0142$ | $0.3725 \pm 0.0132$ | $0.3687 \pm 0.0148$ | $0.3961 \pm 0.0284$ | $0.3730 \pm 0.0145$ | $\mathbf{0.3670} \pm \mathbf{0.0137}$ |
| Photo | GCN | $0.2877 \pm 0.0108$ | $0.2892 \pm 0.0110$ | $0.2913 \pm 0.0122$ | $\mathbf{0.2725} \pm \mathbf{0.0133}$ | $0.3717 \pm 0.0737$ | $0.2867 \pm 0.0113$ | $0.2750 \pm 0.0117$ |
| | GAT | $0.2712 \pm 0.0205$ | $0.2703 \pm 0.0166$ | $0.2692 \pm 0.0161$ | $0.2657 \pm 0.0185$ | $0.3228 \pm 0.0563$ | $0.2704 \pm 0.0172$ | $\mathbf{0.2638} \pm \mathbf{0.0156}$ |
| CS | GCN | $0.2196 \pm 0.0119$ | $0.2142 \pm 0.0056$ | $0.2162 \pm 0.0049$ | $0.2141 \pm 0.0055$ | $0.2778 \pm 0.0583$ | $0.2132 \pm 0.0057$ | $\mathbf{0.2127} \pm \mathbf{0.0054}$ |
| | GAT | $0.2451 \pm 0.0084$ | $0.2432 \pm 0.0057$ | $0.2425 \pm 0.0053$ | $0.2428 \pm 0.0057$ | $0.2786 \pm 0.0350$ | $0.2422 \pm 0.0054$ | $\mathbf{0.2416} \pm \mathbf{0.0051}$ |
| Physics | GCN | $0.1199 \pm 0.0043$ | $0.1190 \pm 0.0035$ | $0.1190 \pm 0.0033$ | $0.1190 \pm 0.0035$ | $0.1289 \pm 0.0114$ | $0.1188 \pm 0.0033$ | $\mathbf{0.1185} \pm \mathbf{0.0034}$ |
| | GAT | $0.1288 \pm 0.0045$ | $0.1287 \pm 0.0043$ | $\mathbf{0.1283} \pm \mathbf{0.0041}$ | $0.1287 \pm 0.0043$ | $0.1334 \pm 0.0055$ | $0.1286 \pm 0.0042$ | $0.1285 \pm 0.0042$ |
| CoraFull | GCN | $1.4310 \pm 0.0221$ | $1.4270 \pm 0.0185$ | $1.4300 \pm 0.0199$ | $1.4210 \pm 0.0182$ | $1.4780 \pm 0.1769$ | $\mathbf{1.4010} \pm \mathbf{0.0189}$ | $1.4070 \pm 0.0120$ |
| | GAT | $1.3670 \pm 0.0217$ | $1.3620 \pm 0.0176$ | $1.3630 \pm 0.0177$ | $1.3610 \pm 0.0175$ | $1.4570 \pm 0.1953$ | $1.3550 \pm 0.0170$ | $\mathbf{1.3490} \pm \mathbf{0.0174}$ |

formulated as follows:

$$\text{NLL} = \frac{1}{N} \sum_{i \in \mathcal{V}}^{N} -y_i \log p_{i,y_i} \tag{15}$$

- **Brier Score** is widely employed metric to quantify the model calibration. It measures the accuracy of model prediction by comparing the predicted probabilities $\mathbf{p}_i$ with the ground-truth occurrences $\mathbf{o}_i$:

$$\text{Brier Score} = \frac{1}{N} \sum_{i \in \mathcal{V}}^{N} \sum_{k=1}^{C} (p_{i,k} - o_{i,k}) \tag{16}$$

Here, $\mathbf{o}_i$ represents a one-hot vector encoding the ground-truth class label.

We report the calibration results assessed by KDE-ECE, class-wise ECE, NLL, and Brier Score in Table 8, 9, 10, and Table 11, respectively. The results demonstrate that our SIMI-MAILBOX generally outperforms state-of-the-art calibration methods on majority of the metrics, including all metrics, particularly when assessed via KDE-ECE, accomplishing state-of-the-art calibration performance across all settings.

### B.5 DIFFERENCE BETWEEN EXISTING GROUPING-BASED CALIBRATIONS

In this subsection, we discuss the key difference between our work and previous categorization-aware calibration approaches (Hébert-Johnson et al., 2018; Perez-Lebel et al., 2022; Yang et al., 2023). While our work shares a similar aspect with prior works in terms of adhering miscalibrations in a group-wise perspective, there exists clear difference between SIMI-MAILBOX and aforementioned works. SIMI-MAILBOX is designed to address the unique calibration challenges posed by the intricate structure of graphs. The methodology and extensive analysis are deeply rooted in the properties of graph data, such as neighborhood affinity, which do not find a direct parallel in the grouping discussed in (Hébert-Johnson et al., 2018; Perez-Lebel et al., 2022; Yang et al., 2023). We will discuss further distinction in one-by-one.

To begin with, the primary distinctions between our work and Hébert-Johnson et al. (2018) lie in:

- **Different Grouping Mechanism**: While Hébert-Johnson et al. (2018) conducts partitioning leveraging the decision tree, our method performs sophisticated categorization based on neighborhood similarity along with confidence levels. The referred work Hébert-Johnson et al. (2018) is not designed to capture neighborhood affinity, which is a pivotal component in GNN calibration domain. Furthermore, it **does not offer the principles for effective**

Table 11: Brier score results (reported in percentage) for our proposed calibration method and baselines, averaged over 75 repetitions (± STD). A lower value indicates better calibration performance.

| Methods | | UnCal. | TS | VS | ETS | CaGCN | GATS | Ours |
|---|---|---|---|---|---|---|---|---|
| Cora | GCN | $0.2828_{\pm 0.0189}$ | $0.2555_{\pm 0.0092}$ | $0.2564_{\pm 0.0096}$ | $0.2555_{\pm 0.0091}$ | $0.2607_{\pm 0.0097}$ | $0.2552_{\pm 0.0100}$ | $\mathbf{0.2541}_{\pm \mathbf{0.0086}}$ |
| | GAT | $0.2766_{\pm 0.0222}$ | $0.2416_{\pm 0.0084}$ | $0.2419_{\pm 0.0105}$ | $0.2416_{\pm 0.0083}$ | $0.2462_{\pm 0.0086}$ | $0.2412_{\pm 0.0080}$ | $\mathbf{0.2402}_{\pm \mathbf{0.0080}}$ |
| Citeseer | GCN | $0.4377_{\pm 0.0494}$ | $0.4094_{\pm 0.0097}$ | $0.4104_{\pm 0.0099}$ | $0.4092_{\pm 0.0099}$ | $0.4157_{\pm 0.0129}$ | $0.4082_{\pm 0.0097}$ | $\mathbf{0.4044}_{\pm \mathbf{0.0084}}$ |
| | GAT | $0.4517_{\pm 0.0508}$ | $0.4099_{\pm 0.0090}$ | $0.4108_{\pm 0.0102}$ | $0.4098_{\pm 0.0090}$ | $0.4107_{\pm 0.0094}$ | $0.4097_{\pm 0.0087}$ | $\mathbf{0.4063}_{\pm \mathbf{0.0078}}$ |
| Pubmed | GCN | $0.2135_{\pm 0.0078}$ | $0.2020_{\pm 0.0039}$ | $0.2024_{\pm 0.0040}$ | $0.2020_{\pm 0.0038}$ | $\mathbf{0.2002}_{\pm \mathbf{0.0039}}$ | $0.2017_{\pm 0.0039}$ | $0.2014_{\pm 0.0038}$ |
| | GAT | $0.2377_{\pm 0.0103}$ | $0.2181_{\pm 0.0042}$ | $0.2178_{\pm 0.0040}$ | $0.2181_{\pm 0.0042}$ | $0.2172_{\pm 0.0042}$ | $0.2180_{\pm 0.0042}$ | $\mathbf{0.2168}_{\pm \mathbf{0.0042}}$ |
| Computers | GCN | $0.1856_{\pm 0.0083}$ | $0.1850_{\pm 0.0073}$ | $0.1842_{\pm 0.0069}$ | $0.1850_{\pm 0.0073}$ | $\mathbf{1.1812}_{\pm \mathbf{0.0074}}$ | $0.1841_{\pm 0.0070}$ | $0.1814_{\pm 0.0068}$ |
| | GAT | $0.1709_{\pm 0.0083}$ | $0.1707_{\pm 0.0080}$ | $0.1692_{\pm 0.0065}$ | $0.1707_{\pm 0.0080}$ | $0.1712_{\pm 0.0078}$ | $0.1708_{\pm 0.0079}$ | $\mathbf{0.1691}_{\pm \mathbf{0.0077}}$ |
| Photo | GCN | $0.1166_{\pm 0.0062}$ | $0.1156_{\pm 0.0067}$ | $0.1157_{\pm 0.0060}$ | $0.1156_{\pm 0.0057}$ | $0.1161_{\pm 0.0049}$ | $0.1151_{\pm 0.0054}$ | $\mathbf{0.1141}_{\pm \mathbf{0.0050}}$ |
| | GAT | $0.1167_{\pm 0.0100}$ | $0.1155_{\pm 0.0079}$ | $0.1143_{\pm 0.0068}$ | $0.1156_{\pm 0.0080}$ | $0.1166_{\pm 0.0072}$ | $0.1156_{\pm 0.0079}$ | $\mathbf{0.1140}_{\pm \mathbf{0.0072}}$ |
| CS | GCN | $0.1032_{\pm 0.0040}$ | $0.1028_{\pm 0.0023}$ | $0.1020_{\pm 0.0020}$ | $0.1018_{\pm 0.0023}$ | $0.1065_{\pm 0.0043}$ | $0.1016_{\pm 0.0024}$ | $\mathbf{0.1014}_{\pm \mathbf{0.0023}}$ |
| | GAT | $0.1133_{\pm 0.0034}$ | $0.1126_{\pm 0.0025}$ | $\mathbf{0.1122}_{\pm \mathbf{0.0023}}$ | $0.1126_{\pm 0.0025}$ | $0.1152_{\pm 0.0037}$ | $0.1126_{\pm 0.0024}$ | $0.1123_{\pm 0.0025}$ |
| Physics | GCN | $0.0614_{\pm 0.0020}$ | $0.0614_{\pm 0.0019}$ | $0.0614_{\pm 0.0018}$ | $0.0614_{\pm 0.0019}$ | $0.0625_{\pm 0.0022}$ | $0.0613_{\pm 0.0019}$ | $\mathbf{0.0612}_{\pm \mathbf{0.0019}}$ |
| | GAT | $0.0657_{\pm 0.0018}$ | $0.0657_{\pm 0.0018}$ | $\mathbf{0.0656}_{\pm \mathbf{0.0018}}$ | $0.0657_{\pm 0.0018}$ | $0.0665_{\pm 0.0018}$ | $\mathbf{0.0656}_{\pm \mathbf{0.0018}}$ | $0.0657_{\pm 0.0018}$ |
| CoraFull | GCN | $0.5231_{\pm 0.0074}$ | $0.5208_{\pm 0.0052}$ | $0.5201_{\pm 0.0050}$ | $0.5207_{\pm 0.0052}$ | $0.5221_{\pm 0.0138}$ | $\mathbf{0.5159}_{\pm \mathbf{0.0054}}$ | $0.5176_{\pm 0.0054}$ |
| | GAT | $0.5117_{\pm 0.0072}$ | $0.5099_{\pm 0.0057}$ | $\mathbf{0.5080}_{\pm \mathbf{0.0057}}$ | $0.5098_{\pm 0.0057}$ | $0.5178_{\pm 0.0162}$ | $0.5089_{\pm 0.0057}$ | $\mathbf{0.5080}_{\pm \mathbf{0.0057}}$ |

Table 12: ECE results (reported in percentage) for our proposed calibration method and GC *with* the holdout set. A lower ECE indicates better calibration performance. Note that **Simi-Mailbox does not have an access** to the holdout data.

| | Datasets | Cora | Citeseer | Pubmed | Computers | Photo | CS | Physics | CoraFull |
|---|---|---|---|---|---|---|---|---|---|
| | GC+TS w/ HO | $3.59_{\pm 1.01}$ | $4.16_{\pm 1.09}$ | $1.27_{\pm 0.31}$ | $3.17_{\pm 0.81}$ | $2.09_{\pm 0.84}$ | $0.99_{\pm 0.20}$ | $0.49_{\pm 0.18}$ | $5.57_{\pm 0.52}$ |
| GCN | GC+ETS w/ HO | $3.29_{\pm 0.94}$ | $3.69_{\pm 1.02}$ | $1.15_{\pm 0.40}$ | $1.45_{\pm 0.45}$ | $1.24_{\pm 0.45}$ | $0.90_{\pm 0.24}$ | $0.48_{\pm 0.20}$ | $4.05_{\pm 0.47}$ |
| | Ours | $\mathbf{2.06}_{\pm \mathbf{0.44}}$ | $\mathbf{2.76}_{\pm \mathbf{0.56}}$ | $\mathbf{0.77}_{\pm \mathbf{0.15}}$ | $\mathbf{1.06}_{\pm \mathbf{0.26}}$ | $\mathbf{1.04}_{\pm \mathbf{0.35}}$ | $\mathbf{0.60}_{\pm \mathbf{0.19}}$ | $\mathbf{0.29}_{\pm \mathbf{0.11}}$ | $\mathbf{3.47}_{\pm \mathbf{1.32}}$ |
| | GC+TS w/ HO | $3.13_{\pm 0.97}$ | $3.85_{\pm 1.16}$ | $1.02_{\pm 0.41}$ | $1.53_{\pm 0.48}$ | $1.63_{\pm 0.79}$ | $0.91_{\pm 0.25}$ | $\mathbf{0.47}_{\pm \mathbf{0.17}}$ | $4.32_{\pm 0.50}$ |
| GAT | GC+ETS w/ HO | $3.15_{\pm 0.98}$ | $3.60_{\pm 1.05}$ | $1.07_{\pm 0.45}$ | $1.26_{\pm 0.37}$ | $1.34_{\pm 0.52}$ | $0.84_{\pm 0.27}$ | $0.51_{\pm 0.23}$ | $3.55_{\pm 0.48}$ |
| | Ours | $\mathbf{2.15}_{\pm \mathbf{0.44}}$ | $\mathbf{2.97}_{\pm \mathbf{0.58}}$ | $\mathbf{0.73}_{\pm \mathbf{0.17}}$ | $\mathbf{0.98}_{\pm \mathbf{0.38}}$ | $\mathbf{1.00}_{\pm \mathbf{0.52}}$ | $0.75_{\pm 0.43}$ | $0.49_{\pm 0.21}$ | $\mathbf{2.66}_{\pm \mathbf{1.01}}$ |

**categorization**, whereas we present the categorization criteria as well based on the novel observation of the correlation between neighborhood affinity and miscalibration level.

- **Methodological Divergence**: While Hébert-Johnson et al. (2018) introduces a universal approach to calibration loss applicable across different domains, SIMI-MAILBOX employs a novel strategy of post-hoc group-specific temperature adjustments that are uniquely suited to the diverging neighborhood affinity in graph data. The grouping scheme in the paper is utilized to **quantify the grouping loss**, rather than developing a new calibration method using the grouping algorithm.

Moreover, there exist key differentiators that set our work apart from Perez-Lebel et al. (2022) as well. Notably, these distinctions include:

- **Unspecified Grouping Mechanism**: While both our method and the approach in Perez-Lebel et al. (2022) emphasize subgroup calibration, our method employs a different grouping mechanism. Our method clusters nodes by assessing similarity in neighborhood predictions and confidence levels within the graph. In contrast, Perez-Lebel et al. (2022) proposes a more generalized framework for multicalibration, but **without** the specific focus on how the **ideal subgraphs** should be generated.

- **Essential Principle of Group-wise Calibration**: The core principle of group-wise calibration is to categorize instances based on **similar degrees of miscalibration**. However, as mentioned in the above, Perez-Lebel et al. (2022) lacks the particular emphasis on this characteristic. Since per-group temperature is uniformly assigned to nodes within the designated group, organizing nodes with diverging levels of miscalibration lead to suboptimal calibration results. In contrast, our method presented straightforward criteria, i.e., neighborhood

Table 13: Average node classification accuracy of various label rate (L/C) for our proposed calibration method and baselines on GCN.

| Datasets | L/C | UnCal. | CaGCN | Ours |
|---|---|---|---|---|
| Cora | 20 | $81.46 \pm 0.29$ | $82.94 \pm 0.19$ | $\mathbf{82.98} \pm 0.37$ |
| | 40 | $83.70 \pm 0.26$ | $84.12 \pm 0.27$ | $\mathbf{84.58} \pm 0.12$ |
| | 60 | $84.40 \pm 0.24$ | $85.54 \pm 0.19$ | $\mathbf{86.06} \pm 0.16$ |
| Citeseer | 20 | $71.64 \pm 0.16$ | $\mathbf{74.90} \pm 0.23$ | $74.44 \pm 0.19$ |
| | 40 | $72.02 \pm 0.26$ | $75.26 \pm 0.36$ | $\mathbf{75.30} \pm 0.32$ |
| | 60 | $73.32 \pm 0.18$ | $76.12 \pm 0.16$ | $\mathbf{76.16} \pm 0.23$ |
| Pubmed | 20 | $79.52 \pm 0.26$ | $81.20 \pm 0.33$ | $\mathbf{81.32} \pm 0.48$ |
| | 40 | $80.42 \pm 0.26$ | $82.78 \pm 0.35$ | $\mathbf{82.82} \pm 0.21$ |
| | 60 | $83.32 \pm 0.15$ | $84.12 \pm 0.28$ | $\mathbf{84.28} \pm 0.40$ |

Table 14: Average node classification accuracy of various label rate (L/C) for our proposed calibration method and baselines on GAT.

| Datasets | L/C | UnCal. | CaGCN | Ours |
|---|---|---|---|---|
| Cora | 20 | $81.78 \pm 0.35$ | $81.98 \pm 0.73$ | $\mathbf{84.14} \pm 0.39$ |
| | 40 | $83.48 \pm 0.36$ | $84.32 \pm 1.08$ | $\mathbf{85.64} \pm 0.45$ |
| | 60 | $84.72 \pm 0.32$ | $85.20 \pm 0.75$ | $\mathbf{86.48} \pm 0.32$ |
| Citeseer | 20 | $70.82 \pm 0.34$ | $73.86 \pm 0.66$ | $\mathbf{74.40} \pm 0.44$ |
| | 40 | $71.64 \pm 0.34$ | $75.28 \pm 0.34$ | $\mathbf{75.82} \pm 0.25$ |
| | 60 | $73.20 \pm 0.21$ | $76.04 \pm 0.37$ | $\mathbf{76.42} \pm 0.13$ |
| Pubmed | 20 | $79.38 \pm 0.35$ | $80.14 \pm 0.36$ | $\mathbf{80.50} \pm 0.24$ |
| | 40 | $80.84 \pm 0.36$ | $82.60 \pm 0.81$ | $\mathbf{82.82} \pm 0.12$ |
| | 60 | $83.42 \pm 0.28$ | $83.36 \pm 0.38$ | $\mathbf{83.78} \pm 0.12$ |

similarity along with confidence level, stemmed from our novel observation of the intimate correlation between nodes sharing similar neighborhood affinity and confidence.

Lastly, our approach possesses different contribution from the proposed method GC in Yang et al. (2023), according to below distinctions:

- **Different Grouping Mechanism**: While both Yang et al. (2023) and SIMI-MAILBOX involve grouping instances for calibration, the underlying principles and mechanisms of these groupings are notably distinct. Our method conducts sophisticated categorization based on neighborhood similarity and confidence levels, specifically tailored to the unique properties of GNNs. Owing to this careful binning principles, leveraging KMeans clustering can lead to effective categorization. In contrast, Yang et al. (2023) proposes a learning-based grouping function that does not explicitly account for the inherent characteristics of GNNs. Although it yields prominent performance in vision domain, the learning-based grouping function, which is in practice *a single linear layer*, is **not sufficient** to capture neighborhood affinity, which will be verified in the subsequent experiment.

- **Methodological Distinction**: In fact, Yang et al. (2023) utilized the **holdout set**, which is partially sampled from the test set, to train the calibration function. However, our experiments demonstrate SIMI-MAILBOX's effectiveness with no need of the holdout set.

To validate the effectiveness of our method and grouping strategy over GC empirically, we conducted additional experiments to compare our SIMI-MAILBOX and Yang et al. (2023) *with* the holdout set (specified as **GC w/ HO**). We adopted two configurations, GC combined with TS and GC with ETS (Zhang et al., 2020), following the combination settings in the original paper. During the evaluation, we randomly sampled 10% of the nodes in the test data and allocated them as the holdout set for GC, adjusting the original evaluation protocol in Yang et al. (2023). Accordingly, we re-evaluate our method in the remaining 90% of the test data. Note that unlike GC, our SIMI-MAILBOX **is trained solely on the validation set**, without an access to the holdout data.

According to the Table 12, our method consistently outperforms all combinations of GC across 15 out of 16 settings, pioneering the effectiveness of proposed grouping strategy over GC, *even* without an access to the holdout set. Further, the results indicate that the learning-based grouping function in GC does not fully encompass the inherent characteristics of GNNs.

### B.6 SIMI-MAILBOX ON SELF-TRAINING

Here, we broaden evaluation of SIMI-MAILBOX on self-training scenarios. We integrated our method into the original CaGCN codebase to maintain consistency. To ensure fair comparisons, we followed the same datasets, split ratio, and evaluation protocols, as well as ensuring the five random seeds uniformly across all experimental setups.

As presented in Table 13 and 14, SIMI-MAILBOX demonstrates superior performance over both uncalibrated GNNs and CaGCN across 17 out of 18 settings in total. This is especially evident in GAT on the Citeseer dataset, where our method achieves a performance increase of 4.18% compared to uncalibrated GAT when L/C=40. These results underscore the efficacy of our method in generating refined pseudo-labels through its sophisticated calibration process.

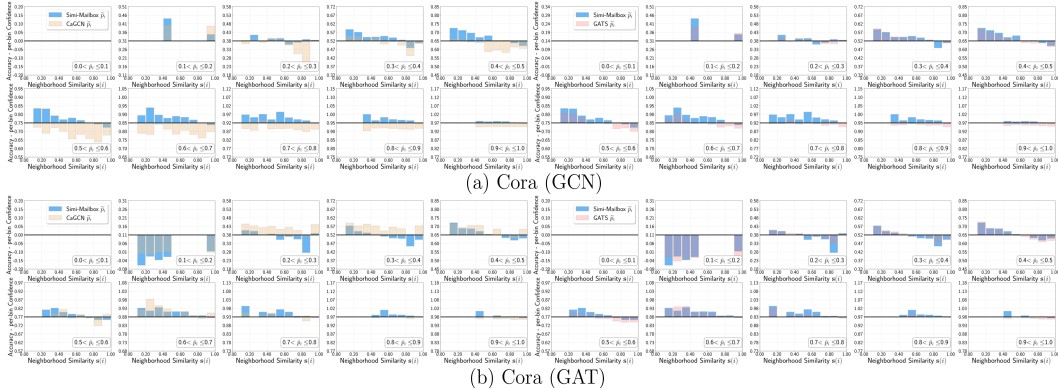

(a) Cora (GCN)

(b) Cora (GAT)

Figure 5: Qualitative analysis of our calibration results on Cora, compared with CaGCN and GATS. The accuracy in per-confidence interval is represented as **black** horizontal lines. The **blue**, **yellow** and **pink** bars illustrate the discrepancy between per-bin accuracy and average confidence of nodes within each affinity sub-intervals, calibrated through our SIMI-MAILBOX, CaGCN, and GATS, respectively.

### B.7 QUALITATIVE ANALYSIS WITH WHOLE BENCHMARK DATASETS

We present additional qualitative comparisons across all benchmark datasets and GNN architecures, as illustrated in Figure 5, 6, 7, 8, 9, 10, 11, 12. Overall, our method achieves better reduction against baselines on the discrepancy between per-confidence accuracy and average confidence varying affinity sub-intervals, notably on Computers and CoraFull.

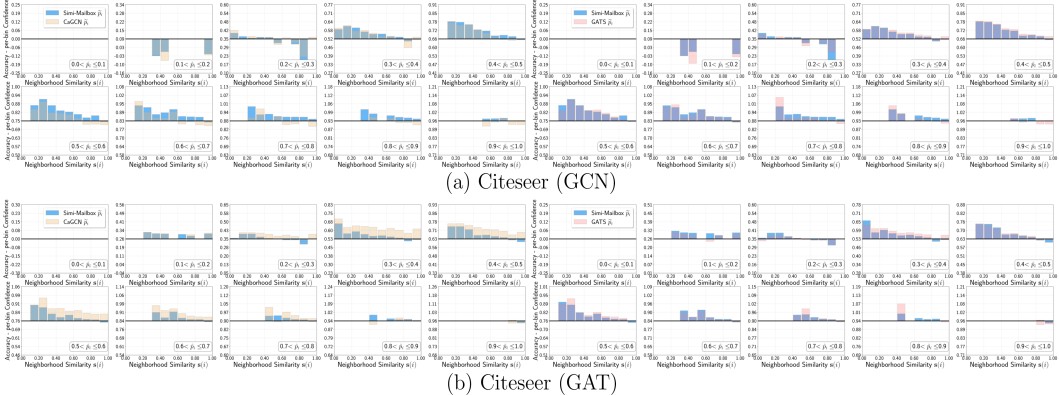

(a) Citeseer (GCN)

(b) Citeseer (GAT)

Figure 6: Qualitative analysis of our calibration results on Citeseer, compared with CaGCN and GATS.

## C FURTHER DISCOVERY OF SURFACE LEARNING IN SECTION 4

To validate the prevalence of our observations on the inconsistency between neighborhood prediction similarity and the foundational assumption in earlier research, we provide further discovery across all benchmark datasets and GNN backbones, depicted in Figure 13, 14, 15, 16, 17, 18, 19, 20. Taking everything into account, it is evident that our discovery is not exclusive to the single case illustrated in Section 4, which breaks down the ground principles of previous GNN calibration studies.

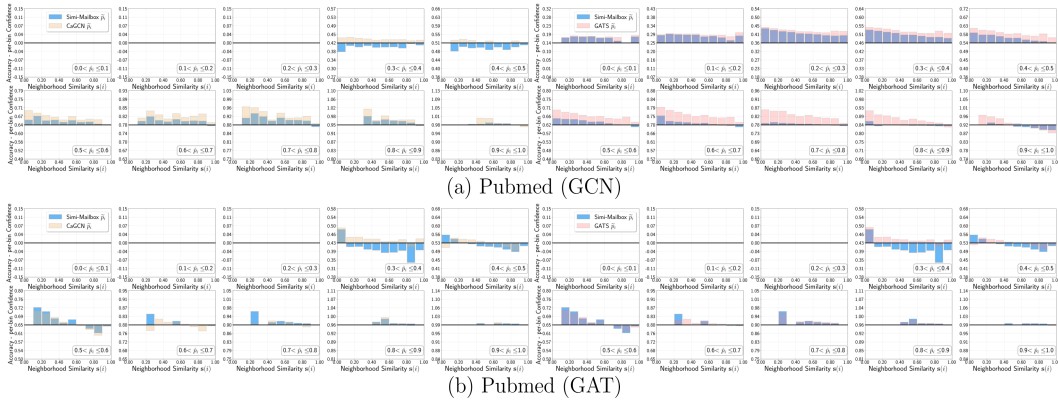

Figure 7: Qualitative analysis of our calibration results on Pubmed, compared with CaGCN and GATS.

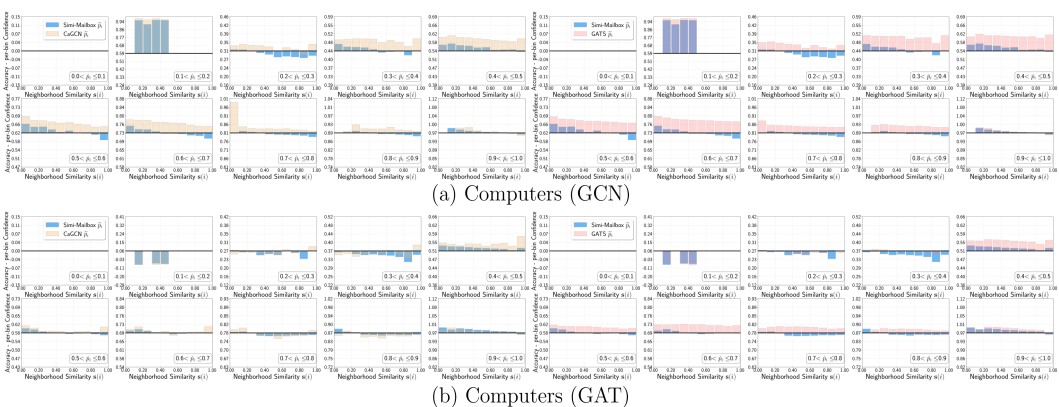

Figure 8: Qualitative analysis of our calibration results on Computers, compared with CaGCN and GATS.

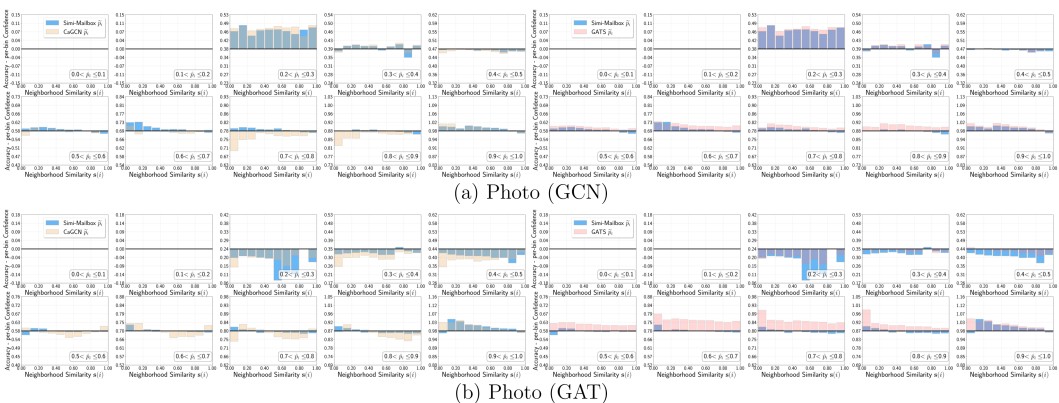

Figure 9: Qualitative analysis of our calibration results on Photo, compared with CaGCN and GATS.

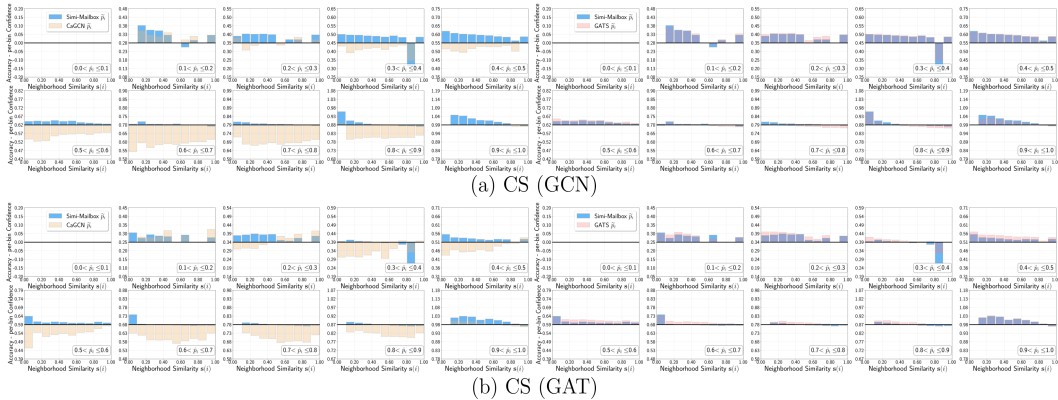

Figure 10: Qualitative analysis of our calibration results on CS, compared with CaGCN and GATS.

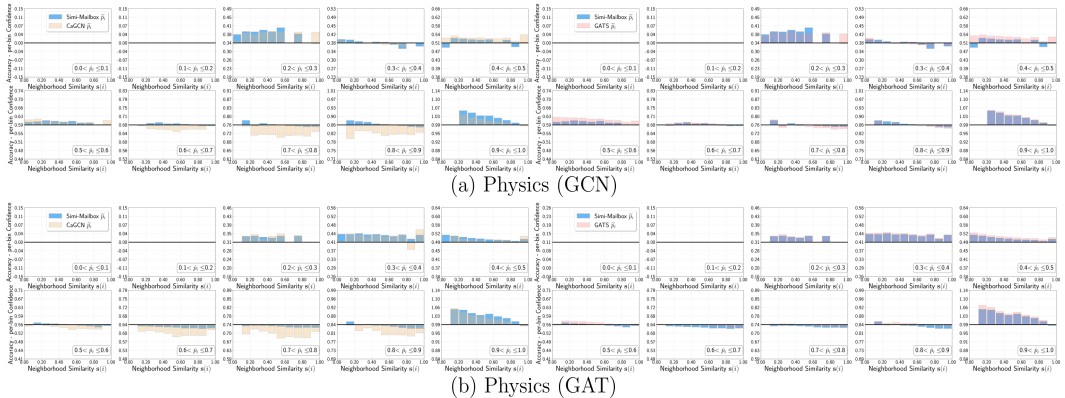

Figure 11: Qualitative analysis of our calibration results on Physics, compared with CaGCN and GATS.

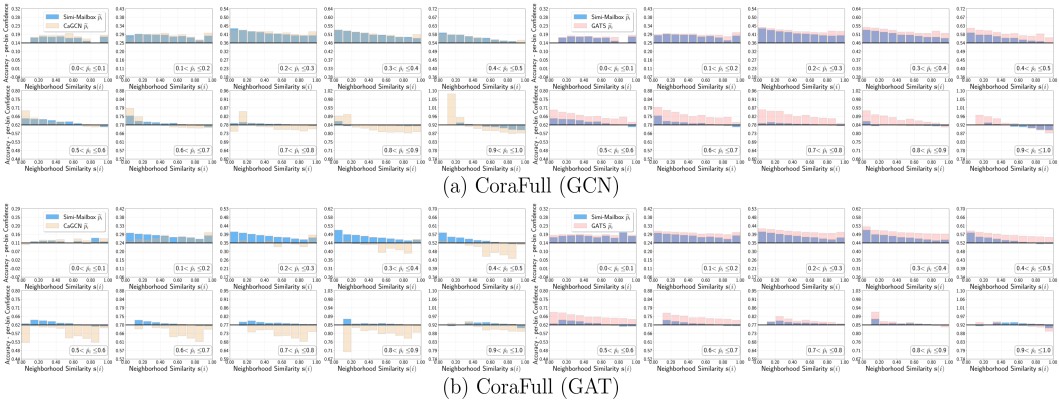

Figure 12: Qualitative analysis of our calibration results on CoraFull, compared with CaGCN and GATS.

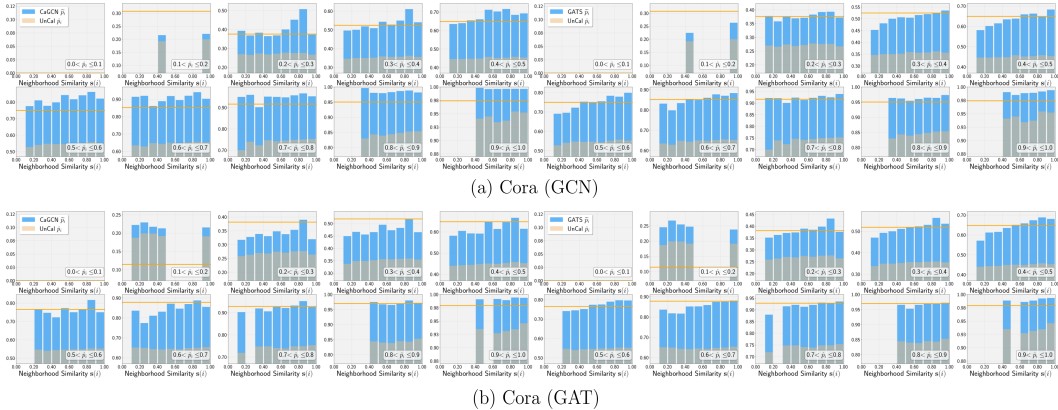

Figure 13: Investigation results of calibrated logits on Cora via CaGCN and GATS. **Orange** line represents *per-confidence* bin accuracy, while **apricot** and **blue** bars denote the average confidence of uncalibrated and calibrated logits in each *neighborhood similarity* sub-interval, respectively. Note that the **gray** area indicates the intersection between two bars.

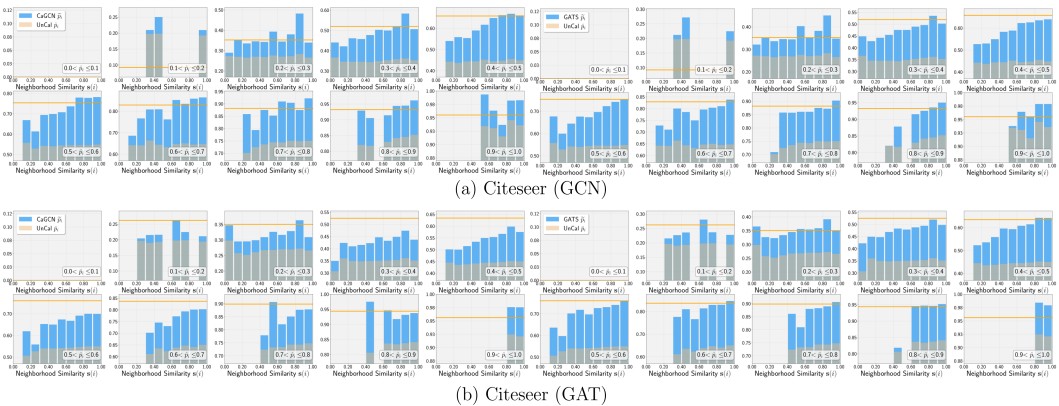

Figure 14: Investigation results of calibrated logits on Citeseer via CaGCN and GATS.

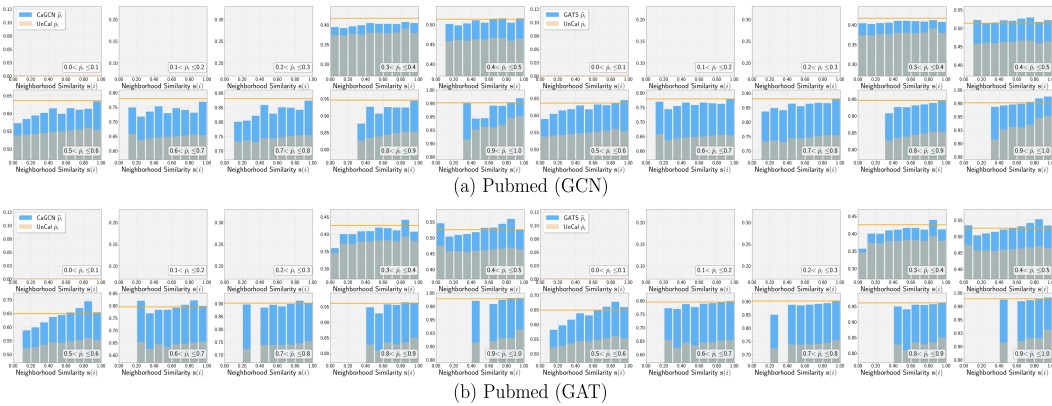

Figure 15: Investigation results of calibrated logits on Pubmed via CaGCN and GATS.

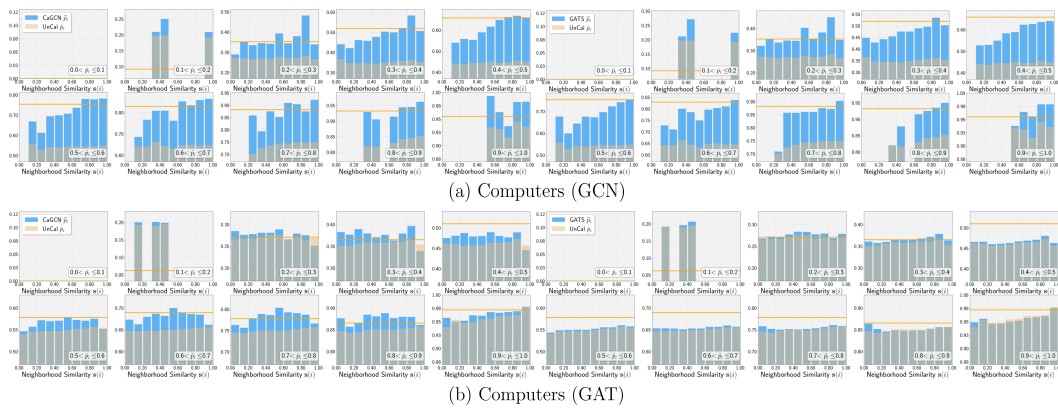

Figure 16: Investigation results of calibrated logits on Computers via CaGCN and GATS.

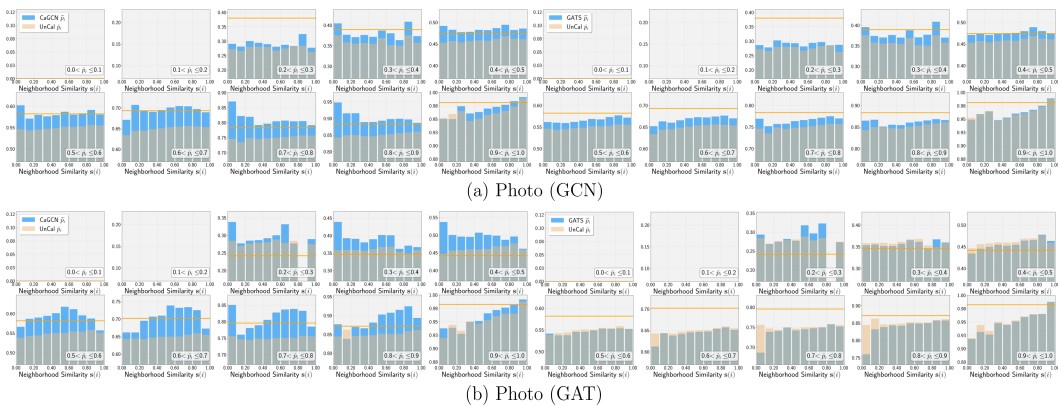

Figure 17: Investigation results of calibrated logits on Photo via CaGCN and GATS.

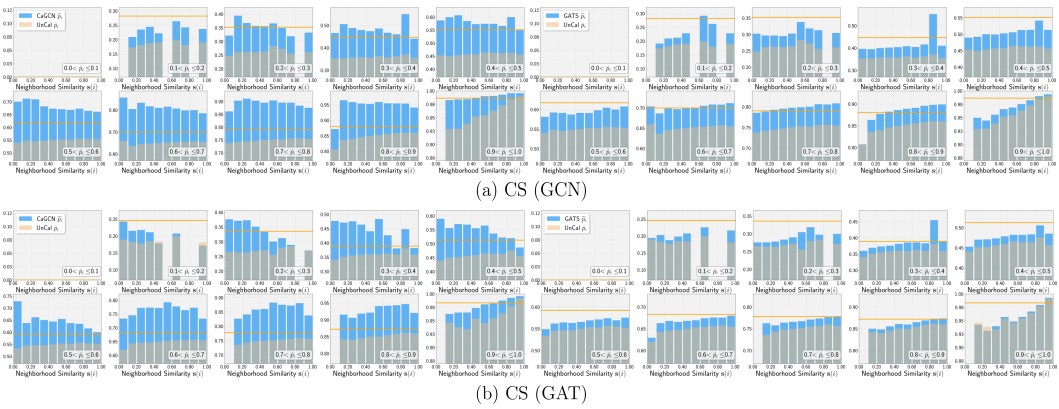

Figure 18: Investigation results of calibrated logits on CS via CaGCN and GATS.

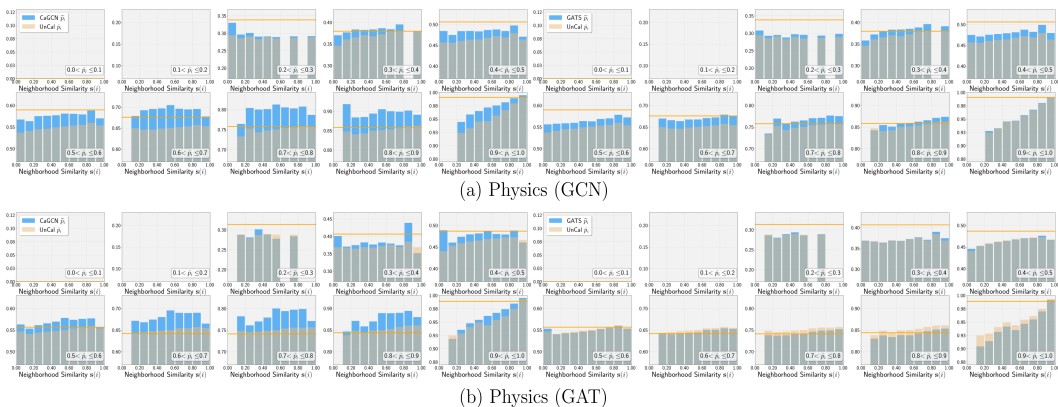

Figure 19: Investigation results of calibrated logits on Physics via CaGCN and GATS.

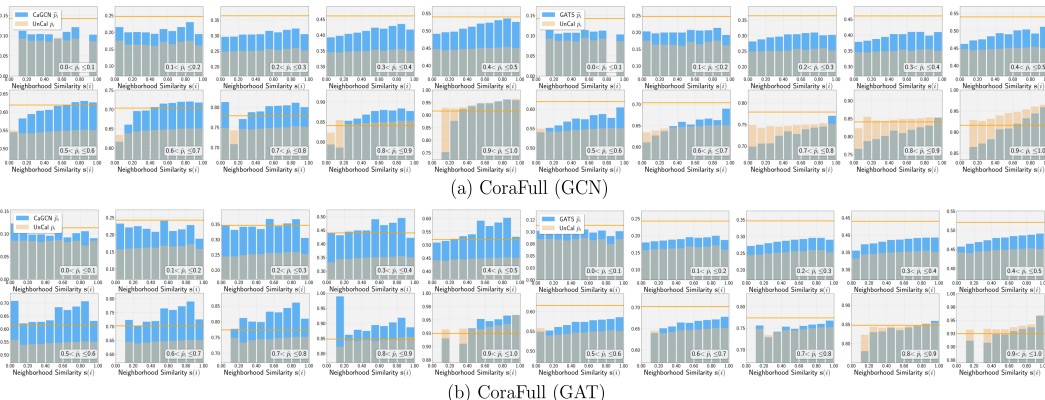

Figure 20: Investigation results of calibrated logits on CoraFull via CaGCN and GATS.

