# OpenReview forum: "Towards Precise Prediction Uncertainty in GNNs: Refining GNNs with Topology-grouping Strategy"
_ICLR.cc/2024/Conference — Submitted to ICLR 2024_

### Official Review · Reviewer_75Vm · 2023-10-29

**Soundness:** 3 good
**Presentation:** 3 good
**Contribution:** 2 fair
**Rating:** 6
**Confidence:** 5

**Summary:**

This paper addresses the issue of graph neural networks (GNNs) calibration by introducing SIMI-MAILBOX, a novel method that categorizes nodes based on neighborhood similarity and confidence levels, effectively mitigating miscalibration across diverse local topologies. The study highlights limitations in current GNN calibration techniques, particularly in relation to neighborhood prediction affinity, and offers valuable insights into improving the reliability of GNN predictions through group-specific temperature adjustments. Experimental validation demonstrates the effectiveness of the proposed approach.

**Strengths:**

1. The paper is well-structured, easy to understand, and presents a straightforward, implementable method.

2. The motivation behind the paper is compelling, and it provides experimental evidence to support the existence of the discussed phenomenon, along with proposing a viable solution.

3. The paper extensively validates the effectiveness of the proposed method through a large number of experiments and visualizations, demonstrating significant improvements compared to baseline approaches.

**Weaknesses:**

1. The paper lacks essential citations to highly relevant works, which significantly diminishes its actual contributions. For instance, [1] introduces a calibration loss based on grouping, and [2] presents the concept of multicalibration, akin to grouping. While these papers do not directly address graphs, they offer universal methods applicable to graphs, and this paper should reference these works, highlighting the similarities and differences between them.

2. The most closely related research is [3], which was first published on June 8, 2023, on Arxiv. In [3], a similar approach of grouping samples and applying temperature scaling to each group is proposed. [3] also raises concerns about potential miscalibration post-grouping. Moreover, [3] defines a general grouping function, and it's worth noting that the grouping in this paper, obtained through k-means clustering, is a specific (human-defined) grouping function. [3] offers a learning-based method in this regard.

3. Based on these two points, the primary contribution of this paper appears to be the discovery of an effective grouping function in the context of GNNs, accompanied by some experimental analysis. While this discovery holds practical significance, its novelty may not be sufficient for acceptance by ICLR.

[1] Alexandre Perez-Lebel, Marine Le Morvan, Gaël Varoquaux. Beyond calibration: estimating the grouping loss of modern neural networks. 2022.

[2] Ursula Hebert-Johnson, Michael Kim, Omer Reingold, Guy Rothblum. Multicalibration: Calibration for the (Computationally-Identifiable) Masses. 2018.

[3] Jia-Qi Yang, De-Chuan Zhan, Le Gan. Beyond Probability Partitions: Calibrating Neural Networks with Semantic Aware Grouping. 2023.

**Questions:**

Please address the weaknesses identified in the review.

---

> ### Author Response · Authors · 2023-11-21
> **Response to Reviewer 75Vm (1/3)**
>
> We sincerely appreciate the reviewer’s helpful and constructive feedbacks. We would like to address the reviewer’s concerns as below.
>
> **[On Comparison with Grouping-based Calibration Methods]**
>
> **W1.** We appreciate the reviewer’s helpful observations and for directing our attention to the work detailed in [1, 2]. We acknowledge the significance of this studies, thereby following the reviewer’s suggestion, we included these studies in the related works section (Section 2) in the revised manuscript.
>
> Nevertheless, our method exhibits **fundamental differences** from [1, 2] in methodology and application. SIMI-MAILBOX is designed to address the unique calibration challenges posed by the intricate structure of graphs. The methodology and extensive analysis are deeply rooted in the properties of graph data, such as neighborhood affinity, which we do not find a direct parallel in the grouping discussed in [1, 2, 3]. We will discuss further distinctions in one-by-one.
>
> To begin with, our work does share the similar aspect with [1] in terms of viewing the calibration status in a group-wise manner. However, the primary distinctions between our work and [1] lie in:
>
> 1. **Different Grouping Mechanism**: While [1] conducts partitioning leveraging decision tree, our method performs sophisticated categorization based on neighborhood similarity along with confidence levels. The referred work [1] is not designed to capture neighborhood affinity, which is a pivotal component in GNN calibration domain. Furthermore, it **does not offer the principles for effective categorization**, whereas we present the categorization criteria based on the novel observation of the correlation between neighborhood affinity and miscalibration level.
> 2. **Methodological Divergence**: While [1] introduces a universal approach to calibration loss applicable across different domains, SIMI-MAILBOX employs a novel strategy of post-hoc group-specific temperature adjustments that are uniquely suited to the diverging neighborhood affinity in graph data. The grouping scheme in [1] is utilized to **quantify the grouping loss**, rather than developing a new calibration method using the grouping algorithm.
>
> Meanwhile, our approach partially aligns with [2] in its objective to address miscalibration among varying subgroups. Nevertheless, there are key differentiators that set our work apart from [2]:
>
> 1. **Unspecified Grouping Mechanism**: While both our method and the approach in [2] emphasize subgroup calibration, SIMI-MAILBOX employs a different grouping mechanism. Our method clusters nodes by assessing similarity in neighborhood predictions and confidence levels within the graph. In contrast, [2] proposes a more generalized framework for multicalibration, but **without** the specific focus on how the **ideal subgraphs** should be generated.
> 2. **Essential Principle of Group-wise Calibration**: The core principle of group-wise calibration is to categorize instances based on **similar degrees of miscalibration**. However, as mentioned in the above, [2] lacks the particular emphasis on this characteristic. Since per-group temperature is uniformly assigned to nodes within the designated group, organizing nodes with diverging levels of miscalibration leads to suboptimal calibration results. In contrast, our method presents straightforward criteria, i.e., neighborhood similarity along with confidence level, stemmed from our novel observation of the intimate correlation between nodes sharing similar neighborhood affinity and confidence.

---

> ### Author Response · Authors · 2023-11-21
> **Response to Reviewer 75Vm (2/3)**
>
> **[On Comparison with GC]**
>
> **W2 & W3.** We appreciate the reviewer’s insightful observation in drawing parallels between our work and the approach presented in [3]. However, we respectfully disagree with the view that our work, SIMI-MAILBOX, is closely related to [3].
>
> While our method shares the group-wise temperature scaling strategy with [3], there exist **clear differences** between SIMI-MAILBOX and the proposed method GC in [3], which we believe are worthwhile to clarify.
>
> 1. **Specialization for GNNs**: Alongside with aforementioned papers [1, 2], the grouping mechanism in [3] does not cover the topological properties of the graph data - a distinction crucial for GNN calibration, leading to suboptimal calibration for addressing GNNs. In contrast, our SIMI-MAILBOX is specifically tailored to the unique challenges and structures of GNNs, thereby accomplishing state-of-the-art performance on diverse benchmark datasets against numerous baselines.
> 2. **Different Grouping Mechanism**: While both [3] and SIMI-MAILBOX involve grouping instances for calibration, the underlying principles and mechanisms of these groupings are notably distinct. Our method conducts sophisticated categorization based on neighborhood similarity and confidence levels, specifically tailored to the unique properties of GNNs. Owing to this careful binning principles, leveraging KMeans clustering can lead to effective categorization. In contrast, [3] proposes a learning-based grouping function that does not explicitly account for the inherent characteristics of GNNs. Although it yields prominent performance in vision domain, the learning-based grouping function, which is in practice *a single linear layer* according to the original code (https://github.com/ThyrixYang/group_calibration/blob/master/methods/group_calibration.py), is **not sufficient to capture neighborhood affinity**, which will be verified in the subsequent experiment.
> 3. **Methodological Distinction**: In fact, [3] utilized the holdout set, which is partially sampled from the **test set**, to train the calibration function. However, our experiments demonstrate SIMI-MAILBOX's effectiveness with no need of the holdout set.
>
> To validate the effectiveness of our method and grouping strategy over [3] empirically, we conducted additional experiments to compare our SIMI-MAILBOX and [3] *with* the holdout set (specified as GC w/ HO). We adopted two configurations, GC combined with TS [4] and GC with ETS [5], following the combination settings in the original paper. Detailed experimental settings is configured in Appendix B.5 in the revised paper. Note that unlike GC, our SIMI-MAILBOX is trained solely on the validation set, without an access to the holdout data.
>
> According to the Table 1 and 2, our method consistently outperforms all combinations of GC across 15 out of 16 settings, pioneering the effectiveness of proposed grouping strategy over GC, **even without** an access to the **holdout set**. For convenience, we attach tables below in Table 12 in the revised paper.
>
> Thus, while there are superficial similarities in using grouping and per-group temperatures for calibration, the specific execution and underlying principles of SIMI-MAILBOX are markedly different from [3], demonstrating predominant calibration performance over GC. Further, the results indicate that the learning-based grouping function in GC does not fully encompass the inherent characteristics of GNNs. We believe these differences are significant and contribute to the distinctiveness and novelty of our work in the domain of GNN calibration.
>
> ---
> **References**
>
> [1] Beyond calibration: estimating the grouping loss of modern neural networks, ICLR 2023.
>
> [2] Multicalibration: Calibration for the (Computationally-Identifiable) Masses, ICML 2018.
>
> [3] Beyond Probability Partitions: Calibrating Neural Networks with Semantic Aware Grouping, NeurIPS 2023.
>
> [4] On Calibration of Modern Neural Networks, ICML 2017.
>
> [5] Mix-n-Match: Ensemble and Compositional Methods for Uncertainty Calibration in Deep Learning, ICML 2020.

---

> > ### Author Response · Authors · 2023-11-21
> > **Response to Reviewer 75Vm (3/3)**
> >
> > Table 1. ECE results (reported in percentage) for our proposed calibration method and GC with the holdout set on GCN. A lower ECE indicates better calibration performance.
> >
> > | GCN          | Cora            | Citeseer        | Pubmed          | Computers       | Photo           | CS              | Physics         | CoraFull        |
> > |--------------|-----------------|-----------------|-----------------|-----------------|-----------------|-----------------|-----------------|-----------------|
> > | GC+TS w/ HO  | 3.59 ± 1.01     | 4.16 ± 1.09     | 1.27 ± 0.31     | 3.17 ± 0.81     | 2.09 ± 0.84     | 0.99 ± 0.20     | 0.49 ± 0.18     | 5.57 ± 0.52     |
> > | GC+ETS w/ HO | 3.29 ± 0.94     | 3.69 ± 1.02     | 1.15 ± 0.40     | 1.45 ± 0.45     | 1.24 ± 0.45     | 0.90 ± 0.24     | 0.48 ± 0.20     | 4.05 ± 0.47     |
> > | **Ours**     | **2.06 ± 0.44** | **2.76 ± 0.56** | **0.77 ± 0.15** | **1.06 ± 0.26** | **1.04 ± 0.35** | **0.60 ± 0.19** | **0.29 ± 0.11** | **3.47 ± 1.32** |
> >
> > ---
> > Table 2. ECE results (reported in percentage) for our proposed calibration method and GC with the holdout set on GAT. A lower ECE indicates better calibration performance.
> >
> > | GAT          | Cora            | Citeseer        | Pubmed          | Computers       | Photo           | CS              | Physics         | CoraFull        |
> > |--------------|-----------------|-----------------|-----------------|-----------------|-----------------|-----------------|-----------------|-----------------|
> > | GC+TS w/ HO  | 3.13 ± 0.97     | 3.85 ± 1.16     | 1.02 ± 0.41     | 1.53 ± 0.48     | 1.63 ± 0.79     | 0.91 ± 0.25     | **0.47 ± 0.17** | 4.32 ± 0.50     |
> > | GC+ETS w/ HO | 3.15 ± 0.98     | 3.60 ± 1.05     | 1.07 ± 0.45     | 1.26 ± 0.37     | 1.34 ± 0.52     | 0.84 ± 0.27     | 0.51 ± 0.23     | 3.55 ± 0.48     |
> > | **Ours**     | **2.15 ± 0.44** | **2.97 ± 0.58** | **0.73 ± 0.17** | **0.98 ± 0.38** | **1.00 ± 0.52** | **0.75 ± 0.43** | 0.49 ± 0.21     | **2.66 ± 1.01** |

---

> > > ### Comment · Reviewer_75Vm · 2023-11-21
> > >
> > > I appreciate the author's diligent effort in addressing my inquiries and conducting relevant experiments. From the author's response, it is evident that the primary contributions of this paper lie in the specialization of GNNs, different grouping mechanisms, and methodological distinctions. This deviates somewhat from the descriptions in the current paper. I hope the author will augment the discussion in the paper to elucidate these differences. Regarding the methodology, the approach in this paper, considering the characteristics of GNNs, outperforms the GC method, which is understandable. I believe adopting distinct grouping strategies for various problems is a reasonable choice. The author's response has effectively addressed most of my concerns, leading me to revise my rating. However, I still contend that this paper's proposed method of grouping and employing different temperatures lacks novelty. Therefore, I will not be upset if this paper is rejected.

---

> > > > ### Author Response · Authors · 2023-11-21
> > > > **Response to Reviewer 75Vm**
> > > >
> > > > We sincerely thank the reviewer 75Vm for the reassessment and improved score of our submission. Your constructive comment has enhanced our work during the rebuttal. In response to the reviewer's helpful suggestion, we have attached the comparative analysis discussed above in the Appendix B.5 in the revised paper. We are glad that our revision and responses have addressed your concerns.

---

### Official Review · Reviewer_jzwr · 2023-11-01

**Soundness:** 3 good
**Presentation:** 3 good
**Contribution:** 2 fair
**Rating:** 6
**Confidence:** 3

**Summary:**

This paper studies the miscalibration issue in graph learning. Different from general deep models, GNNs are often underconfident about their predictions. Previous GNN calibration techniques adapt node-wise temperature scaling by smoothing the confidence of individual nodes with those of adjacent nodes. However, the authors find that calibrated logits from preceding research significantly contradict their foundational assumption of nearby affinity. As a remedy, they introduce SIMI-MAILBOX, which categorizes nodes based on
both neighborhood representational similarity and their own confidence. The extensive experiments verify the effectiveness of their proposed method.

**Strengths:**

[+] The manuscript is well-presented. The authors clearly present the motivations, the methods and the experiments.
[+] The paper is motivated by valid empricial study.
[+] Extensive experiments are performed to verify the effectiveness of the proposed method.

**Weaknesses:**

[-] The pioneering work (CaGCN) [1] in this field (GNN calibration) verify the the Accuracy-Preserving property through theoretical analysis and empirical study. Does the method proposed in this manuscript possess such property? It is an important property for calibration methods. More theoretical justifications and experimental results are expected.
[-] The contribution of the proposed method is somewhat limited. Specifically, the authors utilize KMeans clustering to classify nodes based on analogous neighborhood prediction similarity and their own confidence. And then, analogous nodes within each cluster is rectified via group-specific temperatures. This is incremental compared to previous temperature-tunning methods [1,2].
[-] Many graph calibration methods are built on the concept of neighborhood prediction similarity. Hence, they can achieve superior performance in homophilous graphs. So, can the proposed method work in heterophilous graphs?
[-] The codes for reproducing the results are not provided.
[-] Minor issue: The scarlar (probability) value $p$ in Eq. (1) is bolded while it ($\textbf{p}$) often denotes a vector.

I am happy to raise the ratings providing that the authors can address my concerns during the rebuttal.

[1] Be confident! towards trustworthy graph neural networks via confidence calibration. (NeurIPS 2021)
[2] What makes graph neural networks miscalibrated? (NeurIPS 2022)

**Questions:**

1. Does the method proposed in this manuscript possess the Accuracy-Preserving property?
2. Can the proposed method work in heterophilous graphs?
3. Will the code for reproducing the results be released?

---

> ### Author Response · Authors · 2023-11-21
> **Response to Reviewer jzwr (1/5)**
>
> We sincerely appreciate the reviewer’s helpful and constructive feedbacks. We would like to address the reviewer’s concerns as below.
>
> **[On Accuracy Preserving Property]**
>
> **W1 & Q1.** Our method's algorithmic structure, which utilizes post-hoc group-specific temperatures, ensures that the relative ordering of predictions remains unchanged.
>
> Let us define $f_g:\mathbb R^K\rightarrow \mathbb R^K$ as a calibration function and $z_i=[z_{i1},z_{i2},...,z_{iK}]^{\mathsf T}$ as a logit vector of node $i$. We denote the group-specific temperature for the group node $i$ belongs to as $t_{g_i}$.
>
> Since the group-wise temperature $t_{g_i}$ is **uniformly** applied to **all elements** of $z_i$, the order between elements in the calibrated logit $f_g(z_i)$ remains unchanged when subjected to the softmax operation $\sigma_{\text{sm}}$:
>
> $$
> \begin{equation}
> f_{g}(z_i)=[z_{i1}/t_{g_i},z_{i2}/t_{g_i},...,z_{iK}/t_{g_i}]^{\mathsf T},\quad
> \breve p_i=\sigma_{\text{sm}}\left(f_g(z_i)\right).
> \end{equation}
> $$
>
> Thus, our SIMI-MAILBOX preserves the classification accuracy of the original GNNs, as the softmax function is order-preserving and the scaling by $t_{g_i}$ does not alter the relative ranking of the logit.
>
> Furthermore, we present the tables below to empirically verify that our method preserves the accuracy of uncalibrated GNNs.
>
> ---
> Table 1. Accuracy of uncalibrated GCN and Simi-Mailbox calibrated on GCN.
>
> | GCN          | Cora       | Citeseer   | Pubmed     | Computers  | Photo      | CS         | Physics    | CoraFull   |
> |--------------|------------|------------|------------|------------|------------|------------|------------|------------|
> | UnCal        |82.87 ± 0.75|72.18 ± 0.92|86.40 ± 0.31|88.12 ± 0.60|92.67 ± 0.35|93.32 ± 0.17|95.98 ± 0.15|63.07 ± 0.52|
> | Ours        |82.87 ± 0.75|72.18 ± 0.92|86.40 ± 0.31|88.12 ± 0.60|92.67 ± 0.35|93.32 ± 0.17|95.98 ± 0.15|63.07 ± 0.52|
>
> ---
> Table 2. Accuracy of uncalibrated GAT and Simi-Mailbox calibrated on GAT.
>
> | GAT          | Cora       | Citeseer   | Pubmed     | Computers  | Photo      | CS         | Physics    | CoraFull   |
> |--------------|------------|------------|------------|------------|------------|------------|------------|------------|
> | UnCal        |83.70 ± 0.68|72.13 ± 0.78|85.15 ± 0.34|88.83 ± 0.63|92.63 ± 0.50|92.58 ± 0.21|95.69 ± 0.12|63.55 ± 0.54|
> | Ours        |83.70 ± 0.68|72.13 ± 0.78|85.15 ± 0.34|88.83 ± 0.63|92.63 ± 0.50|92.58 ± 0.21|95.69 ± 0.12|63.55 ± 0.54|

---

> ### Author Response · Authors · 2023-11-21
> **Response to Reviewer jzwr (2/5)**
>
> **[On Limited Contribution]**
>
> **W2.** We thank the reviewer’s constructive comment regarding the nature of our proposed method. While our method shares the post-hoc temperature scaling scheme with previous works like CaGCN [1] and GATS [2], it introduces a novel approach in several key aspects.
>
> To begin with, SIMI-MAILBOX employs **group-specific** temperatures, **regardless of proximity or connectivity**. Our method surpasses the **node-wise** adjustments typical in existing models like CaGCN and GATS, as these models are **highly dependent on adjacent counterparts**. Specifically, CaGCN derives per-node temperatures by leveraging GCN to propagate the logits of nodes to their corresponding neighbors. GATS yields node-wise temperature by propagating GAT to aggregate the neighbors’ logits and distance to training nodes, utilizing attention coefficient to encode neighborhood similarity. Conversely, our approach distinctively leverages KMeans clustering to categorize nodes, taking into account of both their prediction similarity and confidence levels. This enables a more granular and accurate calibration.
>
> Furthermore, the significant innovation of SIMI-MAILBOX lies in our discovery that nodes with similar degrees of neighborhood affinity and confidence exhibit analogous calibration errors. This characteristic, **not explored** in prior works [1, 2], guides our unique categorization strategy and contributes to state-of-the-art calibration performance.
>
> Additionally, our method achieves significant calibration time reduction against [1, 2], as depicted in Table 4. This is especially pronounced in comparison with GATS [2], which is verified to require significantly higher time consumption. This is due to the involvement of extensive parameters having distinct roles in calibration. Besides, GATS requires precomputation of the distance to training data for every node via BFS, which is **computationally prohibitive** when it comes to the large-scale graph data (Note that we did not include the duration of this preprocessing stage in GATS on Table 4). However, our SIMI-MAILBOX can reach markedly faster convergence owing to its straight forward design, without the need for preprocessing.
>
> Thus, while there are some conceptual similarities with existing methods, SIMI-MAILBOX's distinct implementation and the new insight substantiate its novelty and valuable contribution to the field of GNN calibration.

---

> ### Author Response · Authors · 2023-11-21
> **Response to Reviewer jzwr (3/5)**
>
> **[On Experiments on Heterophilous Graphs]**
>
> **W3 & Q3.** In response to the reviewer’s insightful question, we extended our evaluations to include heterophilous graphs, as detailed in below table. Our benchmark datasets for this experiment included Chameleon, Squirrel, Actor, Texas, Wisconsin, and Cornell [3, 4]. We compared SIMI-MAILBOX against uncalibrated GNNs (UnCal), temperature scaling (TS) [6], and GATS. We adopted 10 different train/validation/test splits provided in the official PyTorch Geometric Library [5]. For each split, we conducted 5 random initialization, resulting in 50 runs in total. We maintained the same seeds to our method and baselines, to ensure fair comparison.
>
> As indicated in Table 1, our method surpasses the baselines in 14 out of 16 settings. Notably, on the Texas dataset, SIMI-MAILBOX achieves ECE reduction of 2.65% and 2.98% for GCN and GAT, respectively, compared to uncalibrated results. Conversely, TS and GATS showed limited effectiveness in reducing calibration error in the Texas; GATS in fact increases ECE beyond the uncalibrated results. While the improvements with heterophilous graphs are less pronounced than those observed with homophilous graphs, SIMI-MAILBOX still effectively mitigates miscalibration than previous calibration methods. This is attributed to our method’s careful categorization on the basis of neighborhood similarity and confidence levels. For convenience, we attach tables below in Table 7 in the revised manuscript.
>
> ---
> Table 1. ECE results (reported in percentage) on heterophilous datasets for our proposed calibration method and baselines on GCN, averaged over 50 runs. A lower ECE indicates better calibration performance.
>
> | GCN      | Chameleon       | Squirrel        | Actor           | Texas            | Wisconsin        | Cornell          |
> |----------|-----------------|-----------------|-----------------|------------------|------------------|------------------|
> | UnCal.   | 9.39 ± 1.90     | 7.23 ± 1.46     | **2.69 ± 0.88** | 18.15 ± 4.13     | 17.76 ± 6.97     | 19.17 ± 4.94     |
> | TS       | 9.42 ± 1.94     | 7.18 ± 1.37     | 2.82 ± 0.95     | 18.12 ± 4.95     | 15.41 ± 5.08     | 19.93 ± 5.31     |
> | GATS     | 8.25 ± 2.15     | 6.85 ± 1.09     | 2.82 ± 1.01     | 18.73 ± 4.34     | 15.76 ± 5.33     | 21.60 ± 5.28     |
> | **Ours** | **7.50 ± 1.40** | **5.40 ± 1.04** | 2.73 ± 0.89     | **15.50 ± 4.44** | **15.19 ± 3.58** | **18.66 ± 5.24** |
>
> ---
> Table 2. ECE results (reported in percentage) on heterophilous datasets for our proposed calibration method and baselines on GAT, averaged over 50 runs. A lower ECE indicates better calibration performance.
>
> | GAT      | Chameleon       | Squirrel        | Actor           | Texas            | Wisconsin        | Cornell          |
> |----------|-----------------|-----------------|-----------------|------------------|------------------|------------------|
> | UnCal.   | 7.27 ± 1.43     | 6.42 ± 1.30     | 3.49 ± 1.11     | 18.51 ± 4.47     | 16.12 ± 4.36     | **14.89 ± 6.35** |
> | TS       | 7.23 ± 1.44     | 6.43 ± 1.31     | 3.29 ± 1.15     | 18.62 ± 3.99     | 15.64 ± 3.25     | 16.00 ± 6.72     |
> | GATS     | 7.79 ± 1.95     | 6.66 ± 1.63     | 3.41 ± 1.14     | 18.91 ± 4.49     | 15.16 ± 2.86     | 18.08 ± 6.74     |
> | **Ours** | **6.75 ± 1.73** | **5.45 ± 1.30** | **2.64 ± 0.99** | **15.53 ± 3.48** | **14.90 ± 3.29** | 18.54 ± 6.35     |

---

> ### Author Response · Authors · 2023-11-21
> **Response to Reviewer jzwr (4/5)**
>
> **[On Code Reproduction]**
>
> **W4 & Q4.** We apologize for any confusion caused regarding the accessibility of our code. To clarify, the link to our code repository was provided in the Appendix A.2 of our paper. We understand that this placement may have made it less noticeable. For your convenience, we have included the code as a zip file in the supplementary materials of our submission. However, due to the large memory of pretrained (uncalibrated) GNN model, we have attached the source code of SIMI-MAILBOX and descriptions for running the code, without the trained model. The full contents including pretrained GNN is provided in https://anonymous.4open.science/r/Simi_Mailbox-0816/.
>
> We hope this resolves any difficulties you encountered in accessing our code and aids in your review of our work.

---

> ### Author Response · Authors · 2023-11-21
> **Response to Reviewer jzwr (5/5)**
>
> **[On Eq. 1]**
>
> **W5.** We sincerely appreciate the reviewer’s correction. We modified the typo in Eq. (1) in the revised paper.
>
> ---
> **References**
>
> [1] Be confident! towards trustworthy graph neural networks via confidence calibration, NeurIPS 2021.
>
> [2] What makes graph neural networks miscalibrated?, NeurIPS 2022.
>
> [3] Multi-scale Attributed Node Embedding, IMA Journal of Complex Networks 2021.
>
> [4] Geom-GCN: Geometric Graph Convolutional Networks, ICLR 2020.
>
> [5] Fast graph representation learning with PyTorch Geometric, ICLR 2019 (RLGM Workshop).
>
> [6] On Calibration of Modern Neural Networks, ICML 2017.

---

> ### Comment · Reviewer_jzwr · 2023-11-22
> **Thanks for the response**
>
> Thanks for the authors' detailed response. The pioneering work (CaGCN) [1] in this field (GNN calibration) evaluate their method in the self-training setting. Why you did not follow their experiments?
>
> [1] Be confident! towards trustworthy graph neural networks via confidence calibration. (NeurIPS 2021)

---

> > ### Author Response · Authors · 2023-11-22
> > **Response to Reviewer jzwr**
> >
> > We sincerely thank the reviewer for your valuable feedback. In our study, we followed the evaluation protocol of the current state-of-the-art model, GATS [1], which does not include self-training experiments. Consequently, we did not incorporate such experiments in our method. We recognize this recent query and are initiating experiments; however, the short remaining rebuttal period may limit our capacity to provide detailed results.
> >
> > We appreciate the positive feedback previously expressed by the reviewer, indicating a willingness to raise the ratings. We wonder our response have addressed your prior concerns, and are open to any further questions or concerns.
> >
> > ---
> > **Reference**
> >
> > [1] What makes graph neural networks miscalibrated?, NeurIPS 2022.

---

> ### Author Response · Authors · 2023-11-22
> **Response to Reviewer jzwr**
>
> **[On Self-training Experiment]**
>
> In response to the reviewer’s thoughtful suggestion, we expanded our evaluation on self-training scenarios. We integrated our SIMI-MAILBOX into the original CaGCN [1] codebase to maintain consistency. To ensure fair comparisons, we followed the same datasets, split ratio, and evaluation protocols. Owing to the constraints of the rebuttal period, we present performance results averaged over five runs, ensuring the same random seeds across all experimental setups.
>
> Note that, the performance reported in the original paper could be slightly different from our results. This discrepancy arises due to a requirement in our environment for a higher version of PyTorch (version 1.9.0) to accommodate dgl-cuda11.1, as opposed to the PyTorch version 1.8.1 used in CaGCN.
>
> As presented in the following tables, SIMI-MAILBOX demonstrates superior performance over both uncalibrated GNNs and CaGCN across 17 out of 18 settings. This is especially evident in GAT on the Citeseer dataset, where our method achieves a performance increase of 4.18% compared to uncalibrated GAT when L/C=40. These results underscore the efficacy of SIMI-MAILBOX in generating refined pseudo-labels through its sophisticated calibration process. For your convenience, we have attached the tables below in Table 13 and 14 in the revised manuscript.
>
> ---
> **References**
>
> [1] Be confident! towards trustworthy graph neural networks via confidence calibration, NeurIPS 2021.
>
> ---
> Table 1. Average node classification accuracy of various label rate (L/C) for our proposed calibration method and baselines on GCN.
>
> | GCN  | L/C | UnCal.          | CaGCN            | Ours           |
> |-----------|-----|----------------------|----------------------|----------------------|
> | Cora      | 20  | 81.46 ± 0.29        | 82.94 ± 0.19        | **82.98 ± 0.37**    |
> |           | 40  | 83.70 ± 0.26        | 84.12 ± 0.27        | **84.58 ± 0.12**    |
> |           | 60  | 84.40 ± 0.24        | 85.54 ± 0.19        | **86.06 ± 0.16**    |
> | Citeseer  | 20  | 71.64 ± 0.16        | **74.90 ± 0.23**    | 74.44 ± 0.19        |
> |           | 40  | 72.02 ± 0.26        | 75.26 ± 0.36        | **75.30 ± 0.32**    |
> |           | 60  | 73.32 ± 0.18        | 76.12 ± 0.16        | **76.16 ± 0.23**    |
> | Pubmed    | 20  | 79.52 ± 0.26        | 81.20 ± 0.33        | **81.32 ± 0.48**    |
> |           | 40  | 80.42 ± 0.26        | 82.78 ± 0.35        | **82.82 ± 0.21**    |
> |           | 60  | 83.32 ± 0.15        | 84.12 ± 0.28        | **84.28 ± 0.40**    |
>
> ---
> Table 2. Average node classification accuracy of various label rate (L/C) for our proposed calibration method and baselines on GAT.
>
> | GAT  | L/C | UnCal.         | CaGCN           | Ours            |
> |-----------|-----|----------------------|----------------------|----------------------|
> | Cora      | 20  | 81.78 ± 0.35        | 81.98 ± 0.73        | **84.14 ± 0.39**    |
> |           | 40  | 83.48 ± 0.36        | 84.32 ± 1.08        | **85.64 ± 0.45**    |
> |           | 60  | 84.72 ± 0.32        | 85.20 ± 0.75        | **86.48 ± 0.32**    |
> | Citeseer  | 20  | 70.82 ± 0.34        | 73.86 ± 0.66        | **74.40 ± 0.44**    |
> |           | 40  | 71.64 ± 0.34        | 75.28 ± 0.34        | **75.82 ± 0.25**    |
> |           | 60  | 73.20 ± 0.21        | 76.04 ± 0.37        | **76.42 ± 0.13**    |
> | Pubmed    | 20  | 79.38 ± 0.35        | 80.14 ± 0.36        | **80.50 ± 0.24**    |
> |           | 40  | 80.84 ± 0.36        | 82.60 ± 0.81        | **82.82 ± 0.12**    |
> |           | 60  | 83.42 ± 0.28        | 83.36 ± 0.38        | **83.78 ± 0.12**    |

---

> ### Comment · Reviewer_jzwr · 2023-11-23
> **Thanks for your detailed response**
>
> Thank you for your detailed response. I am glad to hear that the experiments have addressed my concerns. As a result, I have raised the score to 6.  Additionally, I suggest incorporating these new results into the final version.

---

> > ### Author Response · Authors · 2023-11-23
> > **Response to Reviewer jzwr**
> >
> > We are deeply grateful to reviewer jzwr for insightful suggestions and the increased score for our paper. Your feedback has been invaluable in enhancing the quality of our work. Following the reviewer's suggestion, we will include these results in the final version.

---

### Official Review · Reviewer_3Tgz · 2023-11-01

**Soundness:** 3 good
**Presentation:** 3 good
**Contribution:** 2 fair
**Rating:** 6
**Confidence:** 2

**Summary:**

This paper points out the problems with GNN calibration and proposes a new calibration method to improve them. This method focuses on the correlation between the similarity of neighboring nodes and calibration performance, and uses this information to make corrections. Experiments on multiple baselines with multiple data sets show that the proposed method has good performance.

**Strengths:**

- Experimentally finding heuristically that conventional GNN calibration methods have biased performance from node to node and the conditions under which they do so.
- Based on the characteristics found, a method to solve those problems is proposed.
- Experiments have shown that the proposed method is effective for multiple data sets and baselines

**Weaknesses:**

- The problems with conventional methods are heuristic. I don't deny that in itself, and the experimental results suggest that the finding is correct. However, it would be better to have an algorithmic point of view, even if not proven.
- Some parts are not reader friendly, for example, Fig.1 and Fig.3 look like similar tables but show different information.

**Questions:**

The results are likely to change depending on the setting of the number of clusters in K-means, the scaling of $\hat{p}_i$,$\bar{M}^{simi}(i)$ and the setting of $\lambda$ in equation (10). What is the impact of these different settings? If the sensitivity is high and they are not easy to set up, it is not sufficient to say that the proposed method is as effective as claimed. Because of this concern, I withhold my rating.

---

> ### Author Response · Authors · 2023-11-21
> **Response to Reviewer 3Tgz (1/4)**
>
> We sincerely appreciate the reviewer’s helpful and constructive feedbacks. We would like to address the reviewer’s concerns as below.
>
> **[On Algorithmic Point of View on Previous GNN Calibration Methods]**
>
> **W1.** Our work acknowledges the heuristic nature of these conventional approaches and, as the reviewer correctly noted, our empirical results support this assertion. In response to the reviewer’s constructive feedback, we provide the limitation of previous GNN calibration approaches for addressing varying similarity levels in the algorithmic perspective as below.
>
> To begin with, the node-wise temperature $T^{\text{CaGCN}}$ for $l$ layers in CaGCN [1] is defined as below:
>
> $$
> \begin{equation}
> T^{\text{CaGCN}} = \sigma^+(A\sigma_{\text{ReLU}}(...A\sigma_{\text{ReLU}}(AZW^{(1)})W^{(2)}...)W^{(l)})\in\mathbb R^{|\mathcal{V}|},
> \end{equation}
> $$
>
> where $\sigma^+$ and $\sigma_{\text{ReLU}}$ denote softplus and ReLU operation, while $Z$ and $W$ represent logits from trained GNNs and trainable weights, respectively.
>
> The foundational assumption of CaGCN for leveraging GCN as a temperature function is that confidence for nodes linked to agreeing nodes should elevate, while that for nodes with disagreeing neighbors should decrease. They assert that GCN can make the confidence of adjacent nodes similar by propagating the predictions to neighboring counterparts. However, according to the above formulation, the temperature function **does not necessarily yield** higher temperatures for nodes with dissimilar neighbors or lower ones for those with similar neighbors, leading to suboptimal calibration results across diverse neighborhood affinity level. Moreover, our findings in Section 4 indicate that this does not hold true, especially for nodes with dissimilar neighbors.
>
> Meanwhile, from the perspective of individual nodes $i$, the temperature function of GATS [2] $T^{\text{GATS}}_i$ is formulated as:
>
> $$
> \begin{equation}
> T^{\text{GATS}}_i=\frac{1}{H}\sum^H_h\sigma^+(\omega\delta\hat c_i+\sum^{}_j\alpha_i{}_j\gamma_j\tau^h_j) + T_0\in\mathbb R, \quad j\in\mathcal N_i
> \end{equation}
> $$
>
> Here, $H$ and $T_0$ signify the number of attention heads and the initial bias term, respectively, with $\omega$ acting as a learnable coefficient to scale the relative confidence $\delta\hat c_i$ against neighborhood. The scaling factor $\gamma$ is introduced to leverage the distance-to-training-nodes property, and $\tau_j$ refers to the original logits $z_j$ transformed by a linear layer, followed by class-wise sorting within individual nodes' logits.
>
> Recall that GATS demonstrated an increment in calibration error with a decrement in representational similarity, thereby introducing attention coefficient $\alpha_{ij}$ to reflect this. While $\alpha_{ij}$ attempts to capture the affinity between nodes, the model's capacity to discern and appropriately adjust for low similarity levels is limited, since the **complex integration of various factors** may lead to suboptimal temperature adjustments. For instance, the impact of the initial bias term $T_0$ and $\omega\delta\hat c_i$ may obscure the neighborhood similarity associated with nodes, which may not adequately capture the distinct calibration needs of nodes in diverse similarity contexts. Consequently, nodes in low or high similarity contexts might receive suboptimal temperature adjustments.

---

> ### Author Response · Authors · 2023-11-21
> **Response to Reviewer 3Tgz (2/4)**
>
> **[On Unfriendly Part]**
>
> **W2.** Thank you for your valuable feedback regarding the presentation of Figures 1 and 3 in our manuscript. We understand your concern about these figures appearing similar in format, which might lead to confusion. To mitigate this, we will consider adjusting the color schemes or formatting to visually differentiate the figures further in the later version of the paper.

---

> ### Author Response · Authors · 2023-11-21
> **Response to Reviewer 3Tgz (3/4)**
>
> **[Sensitivity Analysis]**
>
> **Q1-1. (On Hyperparameter Robustness)** In response to the the reviewer’s helpful suggestion, we conducted extensive hyperparameter sensitivity analysis for **GCN and GAT** across **all** benchmark datasets. The whole results are depicted in Figure 4 in the revised manuscript. We compare the ECE results of the strongest baseline GATS [2] (specified as dark brown) and ours across varying values of a scaling factor $\lambda$ (specified as green) and the number of bins $N$ (specified as pink) within the range of [5, 10, 15, 20, 25, 30].
>
> As demonstrated in Figure 4, SIMI-MAILBOX consistently outperforms the baseline across all hyperparameter configurations throughout diverse settings, with the number of bins $N$ demonstrating particularly stable performance trends. This robustness is attributed to our method's design, which accounts for the correlation between neighborhood similarity and the degree of miscalibration.
>
> ---
>
> **Q1-2. (Robustness on Scaling Functions)** Regarding the reviewer’s concern with scaling $\hat p_i$ and $\bar{\mathcal M}_{simi}(i)$, the choice of scaling is rooted in the intuition of potential disparity in distributions of neighborhood similarity and confidence. For instance, while neighborhood similarity can be evenly distributed between 0 and 1, confidence in a high-accuracy dataset might be concentrated at higher values. In this situation, min-max scaling is an effective technique for normalizing data, especially when the values are concentrated in a specific range. However, to further enhance the robustness with respect to the design choice of scaling, we conducted additional experiment on our method with **standard scaling** (standard normalization) for constructing a feature vector, illustrated in tables below.
>
> According to Table 1 and 2, SIMI-MAILBOX equipped with standard scaling consistently outperforms the strongest baseline in 15 of the 16 settings. Moreover, the performance gap between our method with standard scaling and the original SIMI-MAILBOX is marginal, suggesting that SIMI-MAILBOX is resilient to different choices of scaling method as well. For convenience, we attach tables below in Table 6 in the revised manuscript.
>
> ---
> **References**
>
> [1] Be confident! towards trustworthy graph neural networks via confidence calibration, NeurIPS 2021.
>
> [2] What makes graph neural networks miscalibrated?, NeurIPS 2022.

---

> ### Author Response · Authors · 2023-11-21
> **Response to Reviewer 3Tgz (4/4)**
>
> Table 1. ECE results (reported in percentage) for our proposed calibration method with min-max scaling and standard scaling, compared to GATS on GCN. A lower ECE indicates better calibration performance.
>
> | GCN                          | Cora               | Citeseer           | Pubmed          | Computers          | Photo              | CS                 | Physics            | CoraFull           |
> |------------------------------|--------------------|--------------------|-----------------|--------------------|--------------------|--------------------|--------------------|--------------------|
> | GATS                         | 3.55 ± 1.28        | 4.49 ± 1.53        | 0.95 ± 0.32     | 2.15 ± 0.52        | 1.46 ± 0.51        | 0.90 ± 0.29        | 0.45 ± 0.15        | 3.74 ± 0.63        |
> | **Ours**                     | **1.97 ± 0.44**    | **2.66 ± 0.53**    | **0.75 ± 0.15** | **1.02 ± 0.26**    | **1.01 ± 0.36**    | **0.58 ± 0.19**    | **0.28 ± 0.11**    | **3.46 ± 1.31**    |
> | **Ours w/ standard scaling** | 2.02 ± 0.54 | 2.67 ± 0.52 | **0.75 ± 0.14** | 1.06 ± 0.24 | 1.01 ± 0.34 | 0.63 ± 0.18 | 0.30 ± 0.10 | 3.53 ± 1.30 |
>
> ---
> Table 2. ECE results (reported in percentage) for our proposed calibration method with min-max scaling and standard scaling, compared to GATS on GAT. A lower ECE indicates better calibration performance.
>
> | GAT                          | Cora               | Citeseer           | Pubmed          | Computers          | Photo              | CS                 | Physics            | CoraFull           |
> |------------------------------|--------------------|--------------------|-----------------|--------------------|--------------------|--------------------|--------------------|--------------------|
> | GATS                         | 3.05 ± 0.78        | 4.01 ± 1.42        | 0.98 ± 0.36     | 1.36 ± 0.34        | 1.49 ± 0.65        | 0.85 ± 0.23        | **0.43 ± 0.16**    | 3.46 ± 0.46        |
> | **Ours**                     | **2.08 ± 0.45**    | **2.86 ± 0.56**    | **0.69 ± 0.16** | 0.95 ± 0.37 | **0.97 ± 0.53**    | **0.72 ± 0.43**    | 0.48 ± 0.22 | **2.64 ± 1.02**    |
> | **Ours w/ standard scaling** | 2.23 ± 0.41 | 2.93 ± 0.58 | **0.69 ± 0.19** | **0.94 ± 0.37**    | 0.98 ± 0.52 | 0.73 ± 0.43 | 0.49 ± 0.21        | 2.82 ± 1.06 |

---

> > ### Comment · Reviewer_3Tgz · 2023-11-21
> >
> > The authors have given me answers that eliminate my concerns regarding my concerns. This raises my score.
> >
> > The similarities to the literature [3], as pointed out by Reviwer 75Vm, are certainly a concern. However, given the nature of this paper, I find it acceptable if it is cited and evaluated, although the novelty may be somewhat impaired.

---

> ### Author Response · Authors · 2023-11-21
> **Response to Reviewer 3Tgz**
>
> We sincerely thank the reviewer 3Tgz for the reassessment and improved score of our submission. Your constructive feedback has enhanced our work during the rebuttal. We are glad that our revision and responses have addressed your concerns.

---

### Author Response · Authors · 2023-11-21
**General Response**

Dear reviewers and AC,

We sincerely appreciate your dedicated time and insightful feedback provided on our manuscript.

As highlighted by the reviewers, we unveil the significant inconsistency between the calibrated logits from preceding GNN calibration methods, particularly concerning their foundational assumptions on neighborhood affinity (3Tgz, jzwr, 75Vm). To address this, we introduce SIMI-MAILBOX, a novel (75Vm) calibration method that categorizes nodes considering both neighborhood similarity and confidence levels. Our method is rooted in our valid empirical study (jzwr) of the correlation between aforementioned factors and the degree of miscalibration (3Tgz). SIMI-MAILBOX demonstrates superior performance over existing state-of-the-art GNN calibration methods under comprehensive experiments (3Tgz, jzwr, 75Vm), encompassing both qualitative and quantitative analysis.

In line with your constructive feedback, we have carefully made the revision with the following additional discussions and experiments:

- The robustness of our SIMI-MAILBOX across varying ranges of two hyperparameters, i.e., a scaling factor $\lambda$ and the number of bins $N$, and different choices of scaling.
- Elucidation of the unique aspects distinguishing our method from contemporary GNN calibration techniques.
- Extensive comparative analysis with other grouping-based calibration approaches.

We have marked the revisions in blue for ease of review. We believe that these enhancements underscore the notable insight of our observation and the effectiveness of SIMI-MAILBOX, offering a valuable contribution to the ICLR community.

We are grateful for your guidance and support throughout this process.

Authors.

---

### Meta-Review · Area_Chair_mNsG · 2023-12-06

**Metareview:**

The paper addresses the issue of calibration in GNNs. GNNs often exhibit miscalibration, where they are underconfident in their predictions. The paper critiques existing GNN calibration methods, which use node-wise temperature scaling, and argues that these methods contradict the foundational assumption of nearby node affinity. To address this issue, the paper introduces a new method, SIMI-MAILBOX, which categorizes nodes based on both neighborhood representational similarity and their own confidence levels. SIMI-MAILBOX involves rectifying analogous nodes within each cluster using group-specific temperatures.

Strengths:
* The paper is well-structured and easy to understand, presenting a straightforward and implementable method.
* The paper is motivated by a valid empirical study that highlights limitations in existing GNN calibration techniques.

Weaknesses:
* The paper lacks citations to highly relevant works diminishing its actual contributions and potential novelty of this study. It should reference relevant works that address calibration and grouping to provide a comprehensive overview of the research landscape.
* The paper's primary contribution appears to be the discovery of an effective grouping function in the context of GNNs. While this discovery is practically significant, its novelty may not be sufficient for acceptance in the conference.

**Justification For Why Not Higher Score:**

The paper addresses an important problem in GNN calibration and proposes a new method that shows promise in improving calibration performance. However, it should address the weaknesses identified in the reviews, particularly in terms of existing literature and novelty, to enhance its contribution.

**Justification For Why Not Lower Score:**

N/A.

---

### Decision · Program_Chairs · 2024-01-16

Reject